# Anderson Accelerated Asynchronous Method for Distributed Optimization

**Cong Li**                                                                *licong@sustech.edu.cn*
*Southern University of Science and Technology*

**Xuyang Wu**[†]                                                           *wuxy6@sustech.edu.cn*
*Southern University of Science and Technology*

**Reviewed on OpenReview:** *https://openreview.net/forum?id=Nm7dTogzfa*

## Abstract

Anderson acceleration (AA) is an effective technique for accelerating fixed-point iterations, but it is rarely applied to distributed optimization. In this paper, we apply AA to accelerate an asynchronous distributed gradient method over the master-worker architecture, resulting in the Asynchronous Distributed Gradient Method with Anderson Acceleration (ADGM-AA). In particular, we first transform the asynchronous gradient method into a fixed-point iteration, and then incorporate it with AA. To ensure the global convergence of ADGM-AA, we equip it with a novel reference-path-based safe-guard scheme. We prove that under mild conditions, ADGM-AA converges with fixed step-sizes that are independent of the delays. Compared with the delay-dependent step-size in most existing works, our delay-free step-size is easier to determine and often leads to faster convergence. Numerical experiments on convex classification tasks show that ADGM-AA improves both iteration-count and wall-clock-time convergence over the baselines in most test examples, while achieving comparable performance in the remaining cases.

## 1 Introduction

This paper focuses on consensus optimization over the master-worker architecture, where one master and $n$ workers collaborate to solve

$$\min_{x \in \mathbb{R}^d} f(x) = \frac{1}{n} \sum_{i=1}^{n} f_i(x). \tag{1}$$

In problem (1), each $f_i : \mathbb{R}^d \to \mathbb{R}$ is a local objective function and is known only by worker $i$. Problem (1) is popular in distributed learning, where each $f_i$ represents a training loss, each worker is a set of GPUs, and multiple GPUs collaborate to train a model.

To scalably solve problem (1), numerous distributed optimization methods are proposed, in which the workers are responsible for most computations, and the master aggregates the computation results from the workers to update the iterate. A critical design choice in distributed methods is the coordination mechanism, leading to two categories of algorithms: synchronous and asynchronous. In synchronous settings, the master and workers must coordinate at each iteration to ensure all workers compute based on a consistent global iterate, which, however, often leads to significant idle time as faster workers wait for stragglers. In contrast, asynchronous algorithms allow the master and workers to proceed independently and update parameters using potentially stale information, which enhances scalability and resource efficiency but usually complicates convergence analysis.

---

[†]Corresponding author.

## 1.1 Related work

In this work, we adapt Anderson acceleration to accelerate an asynchronous distributed optimization method. Consequently, our literature survey focuses on works on Anderson acceleration and asynchronous distributed optimization methods.

### 1.1.1 Anderson acceleration

Anderson acceleration (AA) is a technique for accelerating fixed-point iterations with historical information, and was first proposed by Donald G. Anderson to solve nonlinear integral equations in 1965 (Anderson, 1965). Later, AA is shown to have a quasi-Newton explanation and reduces to the GMRES method (Saad & Schultz, 1986) for solving linear equations (Walker & Ni, 2011). Due to its effectiveness, AA is applied to many research areas such as computational physics (Eyert, 1996), material sciences (Pulay, 1980), and computational chemistry (Pulay, 1982). Although most optimization methods can be formulated into a fixed-point iteration, the application of AA to optimization has occurred only in recent years (Bertrand & Massias, 2021; Li & Li, 2020; Liu et al., 2024; Mai & Johansson, 2020; Ouyang et al., 2020; Scieur et al., 2016). Specifically, the work (Bertrand & Massias, 2021) adapts AA to the coordinate descent method, the works (Li & Li, 2020; Mai & Johansson, 2020; Liu et al., 2024) apply AA to the proximal gradient method, the work (Li & Li, 2020) first incorporates AA with Chebyshev polynomials and then applies the resulting technique to the gradient descent method, the work (Liu et al., 2024) uses AA to accelerate an energy-adaptive gradient method, and the work (Ouyang et al., 2020) applies AA to the alternating direction method of multipliers (ADMM).

Despite the rich literature on AA, existing works only focus on centralized optimization, and the application of AA to distributed optimization remains unexplored.

### 1.1.2 Asynchronous distributed optimization

Due to the excellent scalability and high resource usage efficiency, asynchronous distributed optimization methods have attracted considerable interest, especially in the machine learning community (Aytekin et al., 2016; Cederberg et al., 2025; Chraibi et al., 2024; Liu et al., 2014; Mishchenko et al., 2018; Wu et al., 2023; 2022; Zhang & Kwok, 2014). These methods can be further categorized based on their communication topology: they either follow a master-worker architecture or operate over a decentralized network. Below, we survey works that consider problem (1) over the master-worker architecture, which is more closely related to this paper.

For asynchronous optimization methods over the master-worker architecture, a large body of works consider deterministic-gradient algorithms such as the asynchronous (proximal) incremental aggregated gradient (Aytekin et al., 2016; Feyzmahdavian & Johansson, 2023; Sun et al., 2019; Vanli et al., 2016), the asynchronous coordinate descent method (Liu et al., 2014; Wu et al., 2023; 2022), the asynchronous bundle methods (Cederberg et al., 2025), and the asynchronous alternating direction method of multipliers (Zhang & Kwok, 2014). In parallel, substantial progress has been made in stochastic optimization, such as the asynchronous stochastic gradient descent method (Leblond et al., 2017; Ma et al., 2023; Recht et al., 2011; Zhao & Li, 2016). Due to the asynchronous implementation, information used in the updates is usually delayed. To guarantee the convergence, most existing asynchronous distributed optimization methods (Aytekin et al., 2016; Cederberg et al., 2025; Feyzmahdavian et al., 2014; Feyzmahdavian & Johansson, 2023; Liu et al., 2014; Vanli et al., 2016; Sun et al., 2019; Zhang & Kwok, 2014) assume that delays are bounded and require the algorithm step-size to rely on and decrease with the delay bound, which often leads to overly small step-sizes and slow convergence due to the usually large upper bound of delays. For example, the work (Mishchenko et al., 2018) implements an asynchronous distributed gradient method over a master-worker architecture with 40 nodes and reports the maximum delay of over 1200. Exceptions include some works (Chraibi et al., 2024; Mishchenko et al., 2018; Wu et al., 2023; Cederberg et al., 2025), which converge with fixed step-sizes independent of the delays. More precisely, the seminal work (Mishchenko et al., 2018) proposes an asynchronous proximal gradient algorithm called DAve-RPG, which inspires the remaining methods, including an asynchronous proximal bundle method (Cederberg et al., 2025), a Bregman variant of the DAve-RPG method (Chraibi et al., 2024), and an asynchronous coordinate update method (Wu et al., 2023).

## 1.2 Contribution

We aim to apply the effective AA technique to distributed optimization. In particular, since DAve-RPG converges with fixed delay-independent step-sizes and inspires a series of subsequent works, we choose to accelerate its variant for smooth problems (DAve-G) with AA. The algorithm development faces two primary challenges: first, the straightforward application of AA to DAve-G cannot be implemented in a distributed way; second, to guarantee the global convergence, we need to design a safe-guard scheme, while it is particularly challenging in the asynchronous and distributed setting.

Despite the aforementioned challenges, we incorporate AA into DAve-G in a novel way, which gives an Asynchronous Distributed Gradient Method with Anderson Acceleration (ADGM-AA). Our main contributions include

1) We rewrite DAve-G as a fixed-point iteration in a novel way and follow the idea of AA (rather than directly using it) to accelerate the fixed-point iteration, which allows the resulting algorithm to be implemented in an asynchronous and distributed way.

2) We propose a novel reference-path-based safe-guard scheme that is distributively implementable.

3) We theoretically prove the convergence of ADGM-AA with the safe-guard scheme and fixed step-sizes independent of delays.

The effectiveness of ADGM-AA is demonstrated numerically via classification tasks with real-world datasets.

## 1.3 Outline

The remaining part of this paper is organized as follows. Section 2 introduces Anderson acceleration and the asynchronous distributed method DAve-G. Section 3 develops ADGM-AA by integrating Anderson acceleration with DAve-G, and designs a reference-path-based safe-guard scheme to guarantee the global convergence of the proposed method. Section 4 analyses the convergence of ADGM-AA, and Section 5 presents our numerical experimental results. Finally, Section 6 concludes the paper.

## 1.4 Notation

Throughout this paper, we use $\|\cdot\|$ to denote the Euclidean norm for vectors and the spectral norm for matrices. For any closed convex set $C$, $P_C(\cdot)$ is the projection operator onto $C$. We use $\mathbf{1}$ to represent the all-one vector and $\mathbf{0}$ the all-zero vector. For any nonnegative integer $m$, we define

$$[m] = \{0, 1, \ldots, m-1\}.$$

# 2 Preliminaries: Anderson acceleration and DAve-G

In this section, we introduce Anderson acceleration (AA) and the asynchronous method DAve-G for solving problem (1), which inspires our algorithm development in Section 3.

## 2.1 Anderson acceleration

AA is a technique for accelerating fixed-point iterations of the following form

$$x_{k+1} = \mathrm{T}(x_k) \tag{2}$$

where $\mathrm{T} : \mathbb{R}^d \to \mathbb{R}^d$ is an operator. The update (2) solves the fixed point of T and describes many algorithms in optimization, such as steepest descent (Li & Li, 2020), and ADMM (Ouyang et al., 2020).

The vanilla AA consists of three steps. First, it considers an intermediate iterate of the following form

$$x_{k+\frac{1}{2}} = \sum_{t=0}^{m-1} \gamma_{t,k} x_{k-t},$$

where $m \geq 1$ is a positive integer, and $\gamma_{t,k}$ are real numbers satisfying

$$\sum_{t=0}^{m-1} \gamma_{t,k} = 1.$$

Second, AA chooses the coefficient vector to minimize an approximation of the fixed-point residual norm $\| T(x_{k+\frac{1}{2}}) - x_{k+\frac{1}{2}} \|$, where $T(x_{k+\frac{1}{2}}) - x_{k+\frac{1}{2}}$ is the residual at the intermediate point $x_{k+\frac{1}{2}}$. In particular, for any $k \geq 0$, define $r_k = T(x_k) - x_k$,

$$R_k = (r_k, r_{k-1}, \ldots, r_{k-m+1}) \in \mathbb{R}^{d \times m}. \tag{3}$$

For a candidate coefficient vector $\Gamma = (\gamma_0, \gamma_1, \ldots, \gamma_{m-1})^\top$, the residual can be approximated as

$$T\left(\sum_{t=0}^{m-1} \gamma_t x_{k-t}\right) - \sum_{t=0}^{m-1} \gamma_t x_{k-t} \approx \sum_{t=0}^{m-1} \gamma_t (T(x_{k-t}) - x_{k-t}) = R_k \Gamma, \tag{4}$$

and AA obtains the step-$k$ coefficient vector $\Gamma_k = (\gamma_{0,k}, \gamma_{1,k}, \ldots, \gamma_{m-1,k})^\top$ by solving

$$\Gamma_k \in \arg\ \min_{\Gamma \in \mathbb{R}^m : \Gamma^\top \mathbf{1} = 1} \|R_k \Gamma\|. \tag{5}$$

Note that when T is affine, the approximation in (4) holds with equality. Finally, AA approximates $T(x_{k+\frac{1}{2}})$ to improve the quality of $x_{k+\frac{1}{2}}$:

$$x_{k+1} = \sum_{t=0}^{m-1} \gamma_{t,k}\, T(x_{k-t}), \tag{6}$$

which uses the approximation

$$T(x_{k+\frac{1}{2}}) = T\left(\sum_{t=0}^{m-1} \gamma_{t,k} x_{k-t}\right) \approx \sum_{t=0}^{m-1} \gamma_{t,k}\, T(x_{k-t}). \tag{7}$$

Similar to (4), the approximation in (7) holds with equality when T is affine.

For affine T, AA could converge globally from an arbitrary initial point (Potra & Engler, 2013; Toth & Kelley, 2015; Walker & Ni, 2011). However, for non-affine T, due to the error arising in the approximations (4) and (7), AA could diverge unless the initial point is sufficiently close to the fixed point (Mai & Johansson, 2020). To guarantee global convergence of AA, additional safe-guard schemes are often required.

## 2.2 DAve-G

DAve-G is an asynchronous distributed gradient method for solving problem (1) over the master-worker architecture (Mishchenko et al., 2018). The corresponding synchronous gradient method takes the form

$$x_{k+1} = x_k - \frac{\alpha}{n} \sum_{i=1}^{n} \nabla f_i(x_k), \tag{8}$$

which is mathematically equivalent to the centralized steepest descent method[1]. To perform (8), at each iteration $k$, each worker $i$ computes the gradient $\nabla f_i(x_k)$ and submits it to the master, and the master aggregates the gradients from all workers to update $x_k$. In this synchronous implementation, fast workers need to wait for slow workers, which causes low efficiency of computing resource usage.

As an asynchronous variant of (8), DAve-G updates as

$$x_{k+1} = \frac{1}{n} \sum_{i=1}^{n} (\hat{x}_k^i - \alpha g_k^i), \tag{9}$$

---

[1]DAve-G allows for multiple inner-loop iterations, and here we only consider its single-inner-loop version.

where $\hat{x}_k^i = x_{k-d_k^i}$ and $g_k^i = \nabla f_i(\hat{x}_k^i)$ for a non-negative integer $d_k^i \in [0, k]$ (referred to as delays). If worker $i$ does not send new information before the $k$th update, the master uses the latest stored copy of $\hat{x}_k^i - \alpha g_k^i$ for that worker. In the implementation of DAve-G, the master keeps the current iterate $x$ (we ignore the iteration index for simplicity). Each worker $i$ repeatedly receives $x$ from the master, computes $y^i = x - \alpha \nabla f_i(x)$, and sends it to the master. After the master receives $y^i$ from some (one or more) workers $i$, it performs $x_+ = \frac{1}{n} \sum_{i=1}^n y^i$, where, for each worker $i$, $y^i$ is the most recent $y^i$ the master received from it. This asynchronous update way causes higher computation resource usage than the synchronous method (8), but causes delays in the update.

## 3  Algorithm development

This section applies AA to accelerate DAve-G. To this end, we first adapt the vanilla AA to DAve-G, and then design a novel reference-path-based safe-guard scheme to guarantee its global convergence.

### 3.1  Asynchronous distributed gradient method with Anderson acceleration

To apply AA to DAve-G, we need to rewrite (9) into a fixed-point iteration, which is not straightforward since the left-hand side is of $d$-dimension and the right-hand side involves $n$ iterates of dimension $d$. To address this issue, we rewrite problem (1) as

$$\min \ F(\mathbf{x}) = \sum_{i=1}^n f_i(x^i) \tag{10}$$

$$\text{s. t. } \mathbf{x} \in \mathcal{C}$$

where each $x^i \in \mathbb{R}^d$, $\mathbf{x} = ((x^1)^\top, \ldots, (x^n)^\top)^\top \in \mathbb{R}^{nd}$ and $\mathcal{C} = \{\mathbf{x} \mid x^1 = x^2 = \ldots = x^n\}$. Then, (9) can be viewed as a projected steepest descent step for solving problem (10): letting $\mathbf{x}_{k+1} = ((x_{k+1})^\top, \ldots, (x_{k+1})^\top)^\top$ where $x_{k+1}$ is computed from (9), we have

$$\mathbf{x}_{k+1} = P_{\mathcal{C}}(\hat{\mathbf{x}}_k - \alpha \nabla F(\hat{\mathbf{x}}_k)), \tag{11}$$

where $\hat{\mathbf{x}}_k = ((\hat{x}_k^1)^\top, \ldots, (\hat{x}_k^n)^\top)^\top$, $\hat{x}_k^i = x_{k-d_k^i}$ $(i = 1, 2, \ldots, n)$ and $P_C(\cdot)$ represents the projection operation onto $\mathcal{C}$.

The update (11) is equivalent to the asynchronous fixed-point iteration

$$\mathbf{x}_{k+1} = \mathrm{T}(\hat{\mathbf{x}}_k), \tag{12}$$

where $\mathrm{T}(\mathbf{x}) = P_{\mathcal{C}}(\mathbf{x} - \alpha \nabla F(\mathbf{x}))$. It is easy to verify that the fixed-point set of T coincides with the solution set of problem (10); hence, when $\mathbf{x}^\star = ((x^\star)^\top, \ldots, (x^\star)^\top)^\top$ satisfies $\mathbf{x}^\star = \mathrm{T}(\mathbf{x}^\star)$, the corresponding $x^\star$ is an optimal solution of problem (1). Compared to the centralized update (2), (12) incorporates delayed information due to the asynchronous implementation.

Given the asynchronous fixed-point update (12), we are ready to incorporate AA with DAve-G. Fix $m > 0$ as an integer. For any $k \geq 0$, define

$$\mathbf{r}_k = \mathrm{T}(\hat{\mathbf{x}}_k) - \hat{\mathbf{x}}_k, \ \mathbf{R}_k = (\mathbf{r}_k, \ldots, \mathbf{r}_{k-m+1}), \tag{13}$$

where each $\mathbf{r}_k$ measures the optimality residual at $\hat{\mathbf{x}}_k$. Following (4) – (6), we consider a combination of $\{\hat{\mathbf{x}}_t\}_{t=k-m+1}^k$:

$$\mathbf{x}_{k+\frac{1}{2}} = \sum_{t=0}^{m-1} \gamma_{t,k} \hat{\mathbf{x}}_{k-t}, \tag{14}$$

and minimize an approximate optimality residual at $\mathbf{x}_{k+\frac{1}{2}}$ to determine $\gamma_{t,k}$: define $\Gamma_k = (\gamma_{0,k}, \ldots, \gamma_{m-1,k})^\top$ and solve

$$\Gamma_k \in \arg \ \min_{\Gamma \in \mathbb{R}^m : \Gamma^\top \mathbf{1} = 1} \ \|\mathbf{R}_k \Gamma\|. \tag{15}$$

Finally, we set

$$\mathbf{x}_{k+1}^{\text{AA}} = \sum_{t=0}^{m-1} \gamma_{t,k} \, \text{T}(\hat{\mathbf{x}}_{k-t}), \tag{16}$$

which approximates $T(\mathbf{x}_{k+\frac{1}{2}})$. Although the resulting iterate $\mathbf{x}_{k+1}^{\text{AA}}$ is of the dimension $nd$, we can reduce its dimension to $d$. To see this, note that $\mathbf{x}_{k+1}^{\text{AA}}$ belongs to the set $\mathcal{C}$ so that all its elements are identical. Then, we set

$$x_{k+1}^{\text{AA}} = \frac{1}{n} \sum_{t=0}^{m-1} \gamma_{t,k} \sum_{i=1}^{n} (\hat{x}_{k-t}^i - \alpha g_{k-t}^i), \tag{17}$$

which equals any element of $\mathbf{x}_{k+1}^{\text{AA}}$. In addition, $m$ is often replaced with $m_k = \min\{m, k\}$ in AA.

Setting $x_{k+1} = x_{k+1}^{\text{AA}}$ applies AA to DAve-G, and we refer to the resulting algorithm as the Asynchronous Distributed Gradient Method with Anderson Acceleration (ADGM-AA). Unlike centralized AA, in the asynchronous setting, the gradients $\nabla f_i$ from different workers are evaluated at different delayed points. Therefore, the derivations (13)–(16) are based on the delayed stacked iterates $\{\hat{\mathbf{x}}_{k-t}\}_{t=0}^{m-1}$, with $k$ fixed.

### 3.2 A reference-path-based safe-guard scheme

Even in the centralized setting, steepest descent with AA does not converge globally for non-linear operators, where a counterexample is provided in Mai & Johansson (2020). To guarantee global convergence of ADGM-AA for non-affine operators, we equip it with a safe-guard scheme. To do so, a straightforward idea is to check whether the AA step yields a decreasing objective value, i.e.,

$$f(x_{k+1}) - f(x_k) \leq -C_k \tag{18}$$

for some $C_k \geq 0$. If it holds, then perform the AA step $x_{k+1} = x_{k+1}^{\text{AA}}$, otherwise, execute the DAve-G step $x_{k+1} = x_{k+1}^{\text{DAve}}$ where $x_{k+1}^{\text{DAve}}$ is the left-hand side of (9). However, the condition (18) cannot be checked in a distributed way since the master in general does not have $f_i$'s.

Alternatively, we come up with the following idea: since DAve-G is guaranteed to converge without any safe-guard schemes, as long as the outcomes of the AA-step and the DAve-G step are sufficiently close, the convergence of ADGM-AA may be guaranteed. Inspired by this, we treat the sequence generated by DAve-G as a *reference path*, and restrict the iterate to be close enough to the reference path. Specifically, at each iteration, we check the following condition:

$$\|x_{k+1}^{\text{AA}} - x_{k+1}^{\text{DAve}}\| \leq \eta_k \tag{19}$$

where $\eta_k \geq 0$. If (19) holds, then $x_{k+1} = x_{k+1}^{\text{AA}}$. Otherwise, we set $x_{k+1} = x_{k+1}^{\text{DAve}}$.

Define

$$\tilde{\eta}_k = \begin{cases} \eta_k, & x_{k+1} = x_{k+1}^{\text{AA}}, \\ 0, & \text{otherwise.} \end{cases} \tag{20}$$

It follows that

$$\|x_{k+1} - x_{k+1}^{\text{DAve}}\| \leq \tilde{\eta}_k, \tag{21}$$

and we require $\tilde{\eta}_k$ to be summable to restrict the distance between $x_{k+1}$ and the reference path:

$$\sum_{k=0}^{\infty} \tilde{\eta}_k < +\infty. \tag{22}$$

The condition (22) can be easily satisfied by, e.g.,

$$\eta_k = c(k+1)^{-(1+\epsilon)}, \tag{23}$$

where $c > 0$ and $\epsilon > 0$.

Compared to (18), the safe-guard condition(19) with $\eta_k$ in (23) is much simpler and can be verified in a distributed way. A detailed implementation of ADGM-AA with the safe-guard scheme is presented in Algorithms 1–2. In the master process below, $S_k$ denotes the set of workers from which the master receives new gradient messages at the $k$th update and is not fixed in advance.

---

**Algorithm 1** Master process in ADGM-AA

---

**Input:** storage size $m \geq 1$, initial iterate $x_0$, safe-guard parameters $c > 0$, $\epsilon > 0$.
**Output:** $x_k$.

1: Send $(x_0, 0)$ to all workers.
2: Receive $(\nabla f_i(x_0), 0)$ from each worker $i$ and set $\hat{x}_0^i = x_0, g_0^i = \nabla f_i(x_0)$.
3: Compute $x_1$ by (9), and then send $(x_1, 1)$ to all workers.
4: Set $k = 1$ and $\mathbf{r}_0 = ((x_1 - x_0)^\top, \ldots, (x_1 - x_0)^\top)^\top$.
5: **while** not converge **do**
6:     **wait** until receives $\{(\nabla f_i(x_{t_i}), t_i)\}_{i \in S_k}$ from a set $S_k$ of workers.
7:     **for** $i = 1, \ldots, n$ **do**
8:         **if** $i \in S_k$ **then**
9:             Set $(\hat{x}_k^i, g_k^i) = (x_{t_i}, \nabla f_i(x_{t_i}))$.
10:        **else**
11:             Set $(\hat{x}_k^i, g_k^i) = (\hat{x}_{k-1}^i, g_{k-1}^i)$.
12:        **end if**
13:     **end for**
14:     Compute $x_{k+1}^{\text{DAve}}$ by (9).
15:     Set $\mathbf{r}_k = ((x_{k+1}^{\text{DAve}} - \hat{x}_k^1)^\top, \ldots, (x_{k+1}^{\text{DAve}} - \hat{x}_k^n)^\top)^\top$ and $m_k \leftarrow \min\{m, k\}$.
16:     Compute $\Gamma_k$ by (15).
17:     Compute $x_{k+1}^{\text{AA}}$ by (17).
18:     **if** the safe-guard condition (19) holds **then**
19:         $x_{k+1} \leftarrow x_{k+1}^{\text{AA}}$.
20:     **else**
21:         $x_{k+1} \leftarrow x_{k+1}^{\text{DAve}}$.
22:     **end if**
23:     $k \leftarrow k + 1$.
24:     Send $(x_k, k)$ to workers $i \in S_k$.
25: **end while**

---

**Algorithm 2** Worker $i$ process in ADGM-AA

---

1: **repeat**
2:     Receive $(x, k)$ from the master.
3:     Compute gradient $\nabla f_i(x)$.
4:     Send $(\nabla f_i(x), k)$ to the master.
5: **until** terminated by the master.

---

**Computation, communication, and memory costs.** ADGM-AA follows the same master-worker communication pattern as DAve-G. At the $k$th master update, each worker in $S_k$ sends a $d$-dimensional gradient vector together with its iteration index to the master, and the master sends a $d$-dimensional iterate together with its iteration index back to these workers. Hence, the per-worker memory and communication cost remains $O(d + 1)$. Since the AA step is performed entirely at the master, ADGM-AA incurs essentially no additional worker-side computation compared with DAve-G; the only extra worker-side requirement is storing and communicating a scalar iteration index.

The additional overhead of ADGM-AA comes from the master-side AA step, whose dominant cost is solving problem (15). Let $m_k = \min\{m, k\}$. Following the QR-based AA implementation in (Mai & Johansson, 2020), solving (15) costs at most $O(m_k^2 + ndm_k)$ per iteration, while the corresponding memory cost is $O(ndm_k + m_k^2 + dm_k)$. The total communication cost per iteration is $O(|S_k|(d + 1))$, due to transmitting $(x_k, k)$ from the master to the workers in $S_k$ and receiving their gradient vectors with iteration indices. This is only slightly higher than the $O(|S_k|d)$ communication cost of DAve-G.

## 4 Convergence analysis

This section analyses the convergence of the proposed ADGM-AA. To this end, we assume that each $f_i$ is smooth, problem (1) has at least one optimal solution, and the delays $\{d_k^i\}$ are bounded.

**Assumption 4.1** (convexity and smoothness)**.** *Each function $f_i$ is convex and $L$-smooth, i.e., it is differentiable, and there exists a constant $L > 0$ such that for any $x, y \in \mathbb{R}^d$,*

$$\|\nabla f_i(x) - \nabla f_i(y)\| \leq L\|x - y\|.$$

**Assumption 4.2** (optimal solution existence)**.** *Problem (1) has at least one optimal $x^\star$.*

**Assumption 4.3** (bounded delay)**.** *It holds that $D \overset{\triangle}{=} \max\limits_{k \geq 0} \max\limits_{i \in \{1,\ldots,n\}} d_k^i < +\infty$.*

Assumptions 4.1–4.3 are standard in asynchronous optimization (Aytekin et al., 2016; Wu et al., 2023) to guarantee convergence. Moreover, it is clear that for any optimal solution $x^\star$ of problem (1), $\mathbf{x}^\star = ((x^\star)^\top, \ldots, (x^\star)^\top)^\top$ is a fixed point to T. Under Assumptions 4.1 – 4.2, we derive the following lemma, which will be used in subsequent analysis.

**Lemma 4.1.** *Suppose that Assumptions 4.1 – 4.2 hold. Let*

$$\mathbf{y}_k = P_{\mathcal{C}}(\hat{\mathbf{x}}_k - \alpha \nabla F(\hat{\mathbf{x}}_k)).$$

*If $\alpha \in (0, 1/L]$, then*

$$\|\mathbf{y}_k - \mathbf{x}^\star\|^2 \leq \|\hat{\mathbf{x}}_k - \mathbf{x}^\star\|^2 - 2\alpha(F(\mathbf{y}_k) - F(\mathbf{x}^\star)). \tag{24}$$

*Proof.* See Appendix A.1. □

With Lemma 4.1, we first show the boundedness of the sequence $\{x_k\}$.

**Lemma 4.2.** *Suppose that Assumptions 4.1–4.3 hold. Let $\{x_k\}$ be generated by ADGM-AA with the safeguard scheme. If $\alpha \in (0, \frac{1}{L})$, then for any $k \geq 0$,*

$$\|x_k - x^\star\| \leq \|x_0 - x^\star\| + \eta_{\text{sum}}$$

*where $\eta_{\text{sum}} = \sum\limits_{t=0}^{\infty} \tilde{\eta}_t$.*

*Proof.* See Appendix A.2. □

**Theorem 4.1.** *Suppose that Assumptions 4.1–4.3 hold. Let $\{x_k\}$ be generated by ADGM-AA with the safe-guard scheme. If $\alpha \in (0, \frac{1}{L})$, then for any $k \geq 1$,*

$$\min_{\ell \leq k} f(x_\ell) - f(x^\star) \leq \frac{3(\alpha L + 1)(\|x_0 - x^\star\| + \eta_{\text{sum}})^2}{2\alpha k/(D+1)}. \tag{25}$$

*Proof.* See Appendix A.3. □

Next, we present the convergence rate under the more restrictive strongly convex assumption.

**Assumption 4.4** (strong convexity)**.** *The function $f$ is $\mu$-strongly convex, i.e., for all $x, y \in \mathbb{R}^d$,*

$$f(y) - f(x) \geq \langle \nabla f(x), y - x \rangle + \frac{\mu}{2}\|y - x\|^2.$$

Note that Assumption 4.4 assumes the sum of the $f_i$'s rather than all $f_i$'s to be strongly convex.

**Theorem 4.2.** *Suppose that Assumption 4.4 and all the conditions in Theorem 4.1 hold. Let $\{x_k\}$ be generated by ADGM-AA with the safe-guard scheme. Then, for any $k \geq 0$, it holds that*

$$\|x_k - x^\star\|^2 \leq \rho^{t_k} \|x_0 - x^\star\|^2 + \sum_{t=0}^{t_k-1} \rho^{t_k-t} \theta_t, \tag{26}$$

*where $\rho = \frac{1}{1+\alpha\mu}$, $t_k = \lceil k/(D+1) \rceil$, $\theta_0 = 2\eta_0(\|x_0-x^\star\|+\eta_{\text{sum}})(1+\alpha L)+(\eta_0)^2$, and $\theta_t = \sum_{\ell=(t-1)(D+1)}^{t(D+1)-1} (2\eta_\ell(\|x_0 - x^\star\| + \eta_{\text{sum}})(1 + \alpha L) + (\eta_\ell)^2)$ for all $t \geq 1$. Moreover, equation (26) implies $\lim_{k \to +\infty} x_k = x^\star$.*

*Proof.* See Appendix A.4. □

**Corollary 4.1.** *Suppose that all the conditions in Theorem 4.2 hold. If $\eta_k \leq c(k+1)^{-b}$ for some $c > 0$ and $b > 1$, then*

$$\sum_{t=0}^{t_k-1} \rho^{t_k-t} \theta_t \leq O\left(\frac{1}{(t_k-1)^b}\right). \tag{27}$$

*It also holds that*

$$\|x_k - x^\star\|^2 \leq O\left(\frac{1}{(t_k-1)^b}\right) \tag{28}$$

*Proof.* See Appendix A.5. □

The convergence results show that ADGM-AA converges globally under the reference-path-based safe-guard scheme and preserves the key theoretical advantage of DAve-G: convergence with fixed step-sizes independent of the delay bound. This is nontrivial because ADGM-AA incorporates Anderson acceleration, whose native form is not globally convergent in general. Thus, the theoretical contribution of ADGM-AA is to introduce Anderson acceleration into asynchronous distributed optimization without sacrificing the delay-independent convergence guarantee. As shown in Section 5, this theoretical design also translates into significant empirical speedups over DAve-G in most test examples.

## 5   Numerical experiments

In this section, we test the practical performance of ADGM-AA on distributed training of classification models using logistic regression and least squares. The experiments are implemented in Python with mpi4py and run on a machine with 48 CPU cores. We set the number of workers in problem (1) to $n = 8$ and use one master process and eight worker processes, each mapped to one core. The delays are induced by actual asynchronous communication and process scheduling, rather than artificially specified.

In our experiments, we compare ADGM-AA with two popular asynchronous methods for solving problem (1), including PIAG (Vanli et al., 2016) and DAve-G (Mishchenko et al., 2018). For ADGM-AA, we set different values $m = 6, 11, 16$ to test the effect of the memory size $m$ (i.e., using information from the previous 5, 10, and 15 iterations, excluding the current iterate), and apply the safe-guard scheme with $\eta_k$ in (20) where $c = 10^8$ and $\epsilon = 10^{-8}$. All methods use tuned step-sizes and are initialized at $x_0 = \mathbf{1}$. We use several real-world datasets[2] summarized in Table 1, where the data in each dataset is evenly partitioned across the workers.

---

[2]The datasets A3A, A6A, W1A, W4A, Rcv1, Cifar10, and Covtype are downloaded from `https://www.csie.ntu.edu.tw/~cjlin/libsvmtools/datasets/`. The datasets Madelon and Gisette are downloaded from `https://archive.ics.uci.edu/ml/datasets`. To adapt Cifar-10 to the binary logistic regression setting, we use only the samples from the twoclasses corresponding to labels 0 and 1, and relabel them as $-1$ and 1, respectively. Similarly, labels 1 and 2 in the Covtype dataset are relabeled as 1 and $-1$, respectively.

Table 1: Datasets

| Dataset | A3A | A6A | W1A | W4A | Rcv1 | Cifar-10 | Covtype | Madelon | Gisette |
|---|---|---|---|---|---|---|---|---|---|
| Samples | 3185 | 11220 | 2477 | 7366 | 20242 | 10000 | 581012 | 2000 | 6000 |
| Features | 123 | 123 | 300 | 300 | 47236 | 3072 | 54 | 500 | 5000 |

## 5.1  Logistic regression

The logistic regression corresponds to problem (1) with

$$f_i(x) = \sum_{j=1}^{M_i} \ln\left(1 + \exp\left(-y_{ij}a_{ij}^\top x\right)\right) + \lambda \|x\|^2,$$

where $a_{ij} \in \mathbb{R}^d$ is a training sample, $y_{ij}$ is the corresponding label, $M_i$ is the number of training samples in worker $i$, and $\lambda = 10^{-3}$. For the logistic regression problem, its gradient function is non-affine, so the operator T is non-affine in Section 3.

We plot the gradient norms, defined as $\|\nabla f(x_k)\|$, versus the number of iterations and display the results in Figure 1. From Figure 1, we observe that 1) ADGM-AA significantly converges faster than the two alternative methods DAve-G and PIAG in most cases; 2) As $m$ increases from 6 to 16, the convergence of ADGM-AA becomes faster for most datasets. Additional experimental results are provided in Appendix B. In several logistic-regression experiments, PIAG and DAve-G decrease $\|\nabla f(x_k)\|$ faster in the first few iterations, while ADGM-AA becomes substantially faster only after a transient phase. This behavior is consistent with the residual-based and local nature of Anderson acceleration, as discussed below and further diagnosed in Appendix C.

## 5.2  Least squares

The least-squares problem corresponds to problem (1) with

$$f_i(x) = \|A_i x - b_i\|^2,$$

where each row of matrix $A_i \in \mathbb{R}^{p_i \times d}$ is a training sample, each element of $b_i \in \mathbb{R}^{p_i}$ is the corresponding label, and $p_i$ is the number of training samples in worker $i$. Here, the operator T is affine in Section 3, because the projection operator $P_C$ is linear and the gradient function of the problem is affine.

The experimental results are exhibited in Figure 2. Similar to the logistic regression case, ADGM-AA shows faster empirical convergence than PIAG and DAve-G in iteration count. Moreover, increasing $m$ from 6 to 16 enhances the convergence speed of ADGM-AA. Additional experimental results and diagnostic results are provided in Appendix B and C, respectively.

**Discussion on the observed convergence behavior.** The different behaviors between the logistic-regression experiments in Figure 1 and the least-squares experiments in Figure 2 are consistent with the residual approximation used in AA. ADGM-AA becomes substantially faster only after a transient phase for logistic regression, but the convergence speed does not vary significantly across iterations for least squares. Using the diagnostic viewpoint in Appendix C, the main reason for the difference is whether the fixed-point operator T is affine. For least squares, the objective is quadratic, $\nabla F$ is affine, and $T(\mathbf{x}) = P_{\mathcal{C}}(\mathbf{x} - \alpha \nabla F(\mathbf{x}))$ is affine because $P_{\mathcal{C}}$ is linear. Therefore, under the Anderson constraint $\sum_t \gamma_{t,k} = 1$, the residual model $\mathbf{R}_k \Gamma_k$ is exact up to numerical precision. For logistic regression, the non-affine operator gives rise to higher-order terms in the local AA bound, which can weaken the effect of the predicted Anderson gain during the early iterations. Hence, in least squares, the Anderson residual approximation is essentially exact, while in logistic regression it becomes reliable only after the iterates enter a local regime.

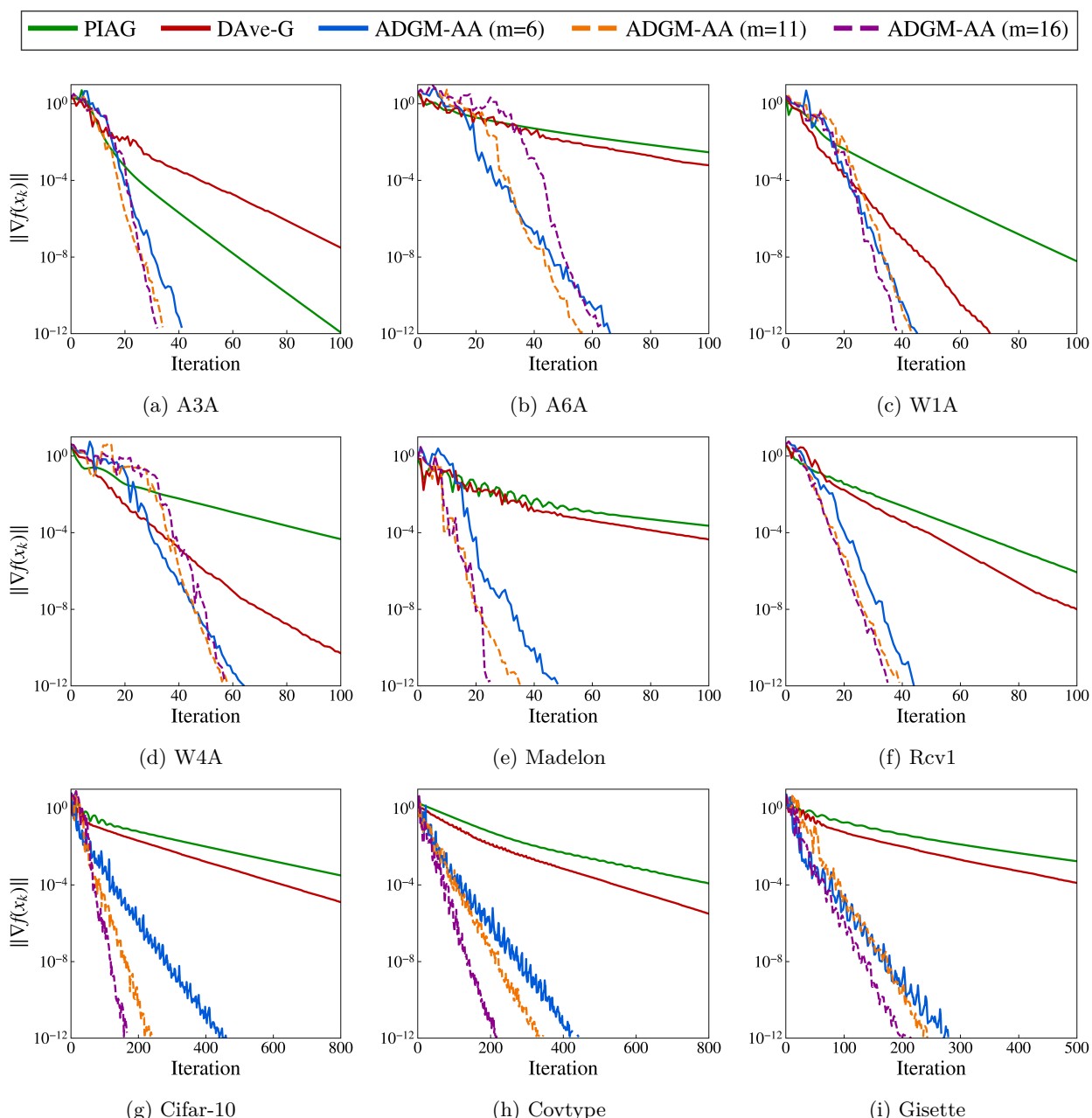

Figure 1: Comparison of PIAG, DAve-G, and ADGM-AA in solving logistic regression.

## 6 Conclusion

We have extended Anderson acceleration (AA) to asynchronous optimization. In particular, we have applied AA to an asynchronous method DAve-G, which faces two key challenges: First, the asynchronous update in DAve-G is not a fixed-point update; Second, it is difficult to design a safe-guard scheme that can be verified in a distributed way. To address these challenges, we first equivalently rewrote DAve-G as a fixed-point iteration by copying variables, and then designed a novel reference-path-based safe-guard condition. We derived the convergence of our method under standard assumptions and a delay-free step-size condition. The experimental results demonstrate that our algorithm shows faster empirical convergence than DAve-G and PIAG.

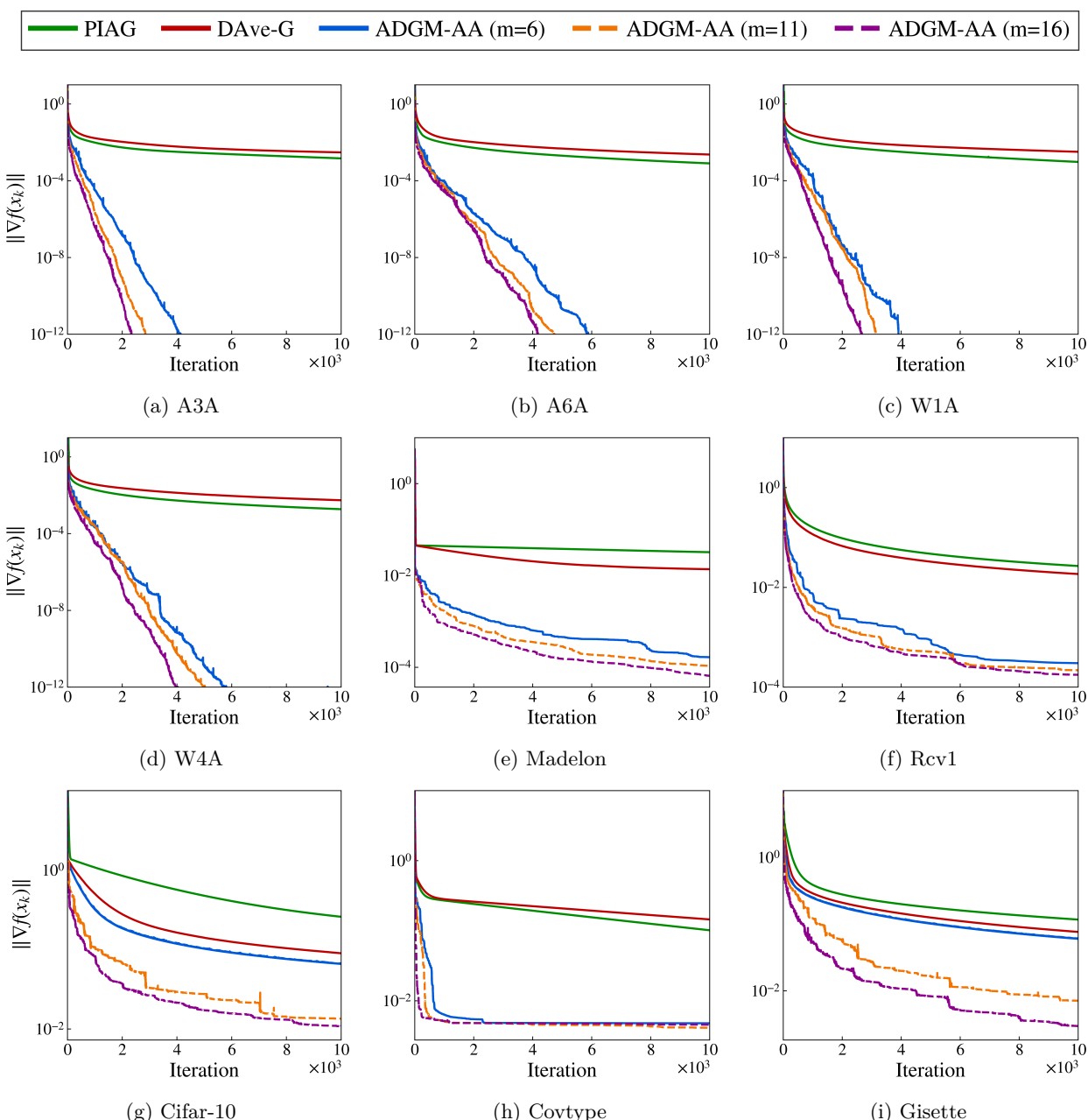

Figure 2: Comparison of PIAG, DAve-G, and ADGM-AA in solving least squares.

## Acknowledgements

We are deeply grateful to Pengyu Zhu for his substantial help throughout the revision of this paper. His insightful feedback on the manuscript and extensive contributions to the numerical experiments played an important role in enhancing both the technical quality and presentation of the work. We also thank the anonymous reviewers for their constructive comments, which helped strengthen the paper. This work is supported in part by the Guangdong Provincial Key Laboratory of Fully Actuated System Control Theory and Technology under Grant No. 2024B1212010002, in part by the Shenzhen Science and Technology Program under Grant No. JCYJ20241202125309014, in part by the Shenzhen Science and Technology Program under Grant No. KQTD20221101093557010, in part by the Guangdong Basic and Applied Basic

Research Foundation under Grant No. 2026A1515012017, and in part by the OpenResearch Project of the State Key Laboratory of Industrial Control Technology under Grant No. ICT2026B102.

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

# A Proofs

## A.1 Proof of Lemma 4.1

Note that $\mathbf{y}_k$ is the result of a projected steepest descent step at $\hat{\mathbf{x}}_k$, and the projected steepest descent method is a special case of the proximal gradient method. Then, applying Wright & Recht, 2022, (9.19) on the proximal gradient method, we have (24).

## A.2 Proof of Lemma 4.2

Define $\beta_k = \max_{t \leq k} \|x_t - x^\star\|$ for all $k \geq 0$. Since

$$\mathbf{y}_k = ((x_{k+1}^{\text{DAve}})^\top, \dots (x_{k+1}^{\text{DAve}})^\top)^\top$$

in Lemma 4.1, we have by (24) that

$$
\begin{aligned}
\|x_{k+1}^{\text{DAve}} - x^\star\|^2 &\leq \frac{1}{n} \sum_{i=1}^n \|\hat{x}_k^i - x^\star\|^2 - 2\alpha(f(x_{k+1}^{\text{DAve}}) - f(x^\star)) \\
&\leq \frac{1}{n} \sum_{i=1}^n \|x_{k-d_k^i} - x^\star\|^2 \leq (\beta_k)^2.
\end{aligned}
\tag{29}
$$

If $x_{k+1} = x_{k+1}^{\text{AA}}$, then by (19) and (21), we have

$$\|x_{k+1} - x^\star\| \leq \|x_{k+1}^{\text{DAve}} - x^\star\| + \|x_{k+1} - x_{k+1}^{\text{DAve}}\| \leq \beta_k + \tilde{\eta}_k. \tag{30}$$

Otherwise,

$$\|x_{k+1} - x^\star\| \leq \|x_{k+1}^{\text{DAve}} - x^\star\| \leq \beta_k.$$

As a result,

$$\beta_{k+1} \leq \beta_k + \tilde{\eta}_k \leq \beta_0 + \sum_{t=0}^\infty \tilde{\eta}_t = \|x_0 - x^\star\| + \eta_{\text{sum}}.$$

## A.3 Proof of Theorem 4.1

The proof consists of two steps. Step 1 proves the following inequality

$$
\begin{aligned}
\|x_{k+1} - x^\star\|^2 &\leq \frac{1}{n} \sum_{i=1}^n \|\hat{x}_k^i - x^\star\|^2 - 2\alpha(f(x_{k+1}) - f(x^\star)) \\
&\quad + \tilde{\eta}_k(\|x_0 - x^\star\| + \eta_{\text{sum}})(2 + 2\alpha L) + (\tilde{\eta}_k)^2,
\end{aligned}
\tag{31}
$$

and Step 2 uses (31) to derive (25).

**Step 1**: If $x_{k+1} = x_{k+1}^{\text{DAve}}$, then by (29), equation (31) naturally holds. Otherwise, by (29) and Lemma 4.2, we have

$$
\begin{aligned}
&\|x_{k+1} - x^\star\|^2 \\
={}& \|x_{k+1}^{\text{DAve}} - x^\star\|^2 + 2\langle x_{k+1}^{\text{AA}} - x_{k+1}^{\text{DAve}}, x_{k+1}^{\text{DAve}} - x^\star\rangle + \|x_{k+1}^{\text{AA}} - x_{k+1}^{\text{DAve}}\|^2 \\
\leq{}& \|x_{k+1}^{\text{DAve}} - x^\star\|^2 + 2\tilde{\eta}_k(\|x_0 - x^\star\| + \eta_{\text{sum}}) + (\tilde{\eta}_k)^2 \\
\leq{}& \frac{1}{n} \sum_{i=1}^n \|\hat{x}_k^i - x^\star\|^2 - 2\alpha(f(x_{k+1}^{\text{DAve}}) - f(x^\star)) + 2\tilde{\eta}_k(\|x_0 - x^\star\| + \eta_{\text{sum}}) + (\tilde{\eta}_k)^2.
\end{aligned}
\tag{32}
$$

Moreover, by the convexity and the $L$-smoothness of $f$,

$$
\begin{aligned}
f(x_{k+1}) - f(x_{k+1}^{\text{DAve}}) &\leq \langle \nabla f(x_{k+1}), x_{k+1} - x_{k+1}^{\text{DAve}}\rangle \\
&= \langle \nabla f(x_{k+1}) - \nabla f(x^\star), x_{k+1} - x_{k+1}^{\text{DAve}}\rangle \\
&\leq L\|x_{k+1} - x^\star\| \cdot \|x_{k+1} - x_{k+1}^{\text{DAve}}\| \\
&\leq L(\|x_0 - x^\star\| + \eta_{\text{sum}})\tilde{\eta}_k,
\end{aligned}
\tag{33}
$$

where the last step uses Lemma 4.2 and (19). Substituting (33) into (32) gives (31).

**Step 2**: Define $\mathcal{I}_0 = \{0\}$ and $\mathcal{I}_t = \{(t-1)(D+1)+1, \ldots, t(D+1)\}$ for each $t \geq 1$. Also let $a_t = \max_{k \in \mathcal{I}_t} \|x_k - x^\star\|^2$. First, by induction we prove that for any $k+1 \in \mathcal{I}_{t+1}$,

$$\|x_{k+1} - x^\star\|^2 \leq a_t - 2\alpha(f(x_{k+1}) - f(x^\star)) + \sum_{\ell=t(D+1)}^{k} b_\ell \tag{34}$$

where $b_\ell = 2\tilde{\eta}_\ell(\|x_0 - x^\star\| + \eta_{\mathrm{sum}})(1 + \alpha L) + (\tilde{\eta}_\ell)^2$. Fix $k = t(D+1)$. By Assumption 4.3, $d_k^i \leq D$ and $k - d_k^i \in \mathcal{I}_t$. Then, by (31) and $\hat{x}_k^i = x_{k-d_k^i}$, we have (34).

Suppose that for some $K+1 \in \mathcal{I}_{t+1}$, equation (34) holds for all $k+1 \leq K$ and $k+1 \in \mathcal{I}_{t+1}$. Then, by (31) and $\hat{x}_K^i = x_{K-d_K^i}$,

$$\|x_{K+1} - x^\star\|^2 \leq \max_{1 \leq i \leq n} \|x_{K-d_K^i} - x^\star\|^2 - 2\alpha(f(x_{K+1}) - f(x^\star)) + b_K. \tag{35}$$

Moreover, $K - d_K^i \in \mathcal{I}_t \cup \{t(D+1)+1, \ldots, K\}$. Then, if $K - d_K^i \geq t(D+1)+1$, by (34), we have that

$$\|x_{K-d_K^i} - x^\star\|^2 \leq a_t + \sum_{\ell=t(D+1)}^{K-d_K^i - 1} b_\ell \leq a_t + \sum_{\ell=t(D+1)}^{K-1} b_\ell.$$

If $K - d_K^i \in \mathcal{I}_t$, we obtain that

$$\|x_{K-d_K^i} - x^\star\|^2 \leq a_t \leq a_t + \sum_{\ell=t(D+1)}^{K-1} b_\ell.$$

Thus, we have that

$$\|x_{K-d_K^i} - x^\star\|^2 \leq a_t + \sum_{\ell=t(D+1)}^{K-1} b_\ell. \tag{36}$$

Substituting the above equation into (35) yields (34) with $k = K$. Concluding all the above, we obtain (34) for all $K+1 \in \mathcal{I}_{t+1}$.

Next, by (34), we have that for all $t \geq 0$,

$$a_{t+1} \leq a_t - 2\alpha \min_{k' \in \mathcal{I}_{t+1}} (f(x_{k'}) - f(x^\star)) + \sum_{\ell=t(D+1)}^{(t+1)(D+1)-1} b_\ell$$

$$\leq a_0 - \sum_{\ell=0}^{t} 2\alpha \min_{k' \in \mathcal{I}_{\ell+1}} (f(x_{k'}) - f(x^\star)) + \sum_{\ell=0}^{(t+1)(D+1)-1} b_\ell, \tag{37}$$

where the second step uses telescoping cancellation. Moreover, for any $k \geq 0$, we have

$$k \in \mathcal{I}_{t_k}, \tag{38}$$

where $t_k = \lceil k/(D+1) \rceil$ and $t_k(D+1) \geq k$. Letting $t = t_k - 1$ in (34) and using (37) gives

$$\|x_k - x^\star\|^2 \leq a_t - 2\alpha(f(x_k) - f(x^\star)) + \sum_{\ell=t(D+1)}^{k-1} b_\ell$$

$$\leq a_t - 2\alpha \min_{k' \in \mathcal{I}_{t_k}} (f(x_{k'}) - f(x^\star)) + \sum_{\ell=t(D+1)}^{t_k(D+1)-1} b_\ell \tag{39}$$

$$\leq a_0 - 2\alpha \big( \min_{k' \in \mathcal{I}_{t_k}} (f(x_{k'}) - f(x^\star)) + \sum_{\ell=0}^{t-1} \min_{k' \in \mathcal{I}_{\ell+1}} (f(x_{k'}) - f(x^\star)) \big) + \sum_{\ell=0}^{t_k(D+1)-1} b_\ell.$$

Then, by the above inequality, we have

$$\min_{\ell \leq k} f(x_\ell) - f(x^\star) \leq \frac{1}{t+1} \left( \min_{k' \in \mathcal{I}_{t_k}} (f(x_{k'}) - f(x^\star)) + \sum_{\ell=0}^{t-1} \min_{k' \in \mathcal{I}_{\ell+1}} (f(x_{k'}) - f(x^\star)) \right)$$

$$\leq \frac{a_0 + \sum\limits_{\ell=0}^{\infty} b_\ell}{2\alpha t_k} \tag{40}$$

$$\leq \frac{a_0 + \sum\limits_{\ell=0}^{\infty} b_\ell}{2\alpha k/(D+1)}$$

Moreover, since

$$\sum_{\ell=0}^{\infty} (\tilde{\eta}_\ell)^2 \leq \left( \sum_{\ell=0}^{\infty} \tilde{\eta}_\ell \right)^2 = \eta_{\text{sum}}^2,$$

we have

$$\sum_{\ell=0}^{\infty} b_\ell \leq 2\eta_{\text{sum}}(\|x_0 - x^\star\| + \eta_{\text{sum}})(1 + \alpha L) + \eta_{\text{sum}}^2. \tag{41}$$

Therefore,

$$a_0 + \sum_{\ell=0}^{\infty} b_\ell \leq \|x_0 - x^\star\|^2 + 2\eta_{\text{sum}}(\|x_0 - x^\star\| + \eta_{\text{sum}})(1 + \alpha L) + \eta_{\text{sum}}^2$$

$$\leq (3\alpha L + 3)(\|x_0 - x^\star\| + \eta_{\text{sum}})^2,$$

substituting which into (40) gives (25).

### A.4   Proof of Theorem 4.2

Under the strong convexity assumption,

$$f(x_{k+1}) - f(x^\star) \geq \frac{\mu}{2} \|x_{k+1} - x^\star\|^2, \tag{42}$$

substituting which into (34) yields

$$(1 + \alpha\mu)\|x_{k+1} - x^\star\|^2 \leq a_t + \sum_{\ell=t(D+1)}^{k} b_\ell \leq a_t + \sum_{\ell=t(D+1)}^{(t+1)(D+1)-1} b_\ell \tag{43}$$

for all $k + 1 \in \mathcal{I}_{t+1}$. By the above inequality,

$$a_{t+1} \leq \rho a_t + \rho \theta_t \leq \rho^{t+1} a_0 + \sum_{\ell=0}^{t} \rho^{t+1-\ell} \theta_\ell. \tag{44}$$

Moreover, for any $k \geq 0$, by (38), we have

$$\|x_k - x^\star\|^2 \leq a_{t_k},$$

which, together with (44), gives (26).

Next, due to (26) and (41) and according to Polyak, 1987, Lemma 3, Sec 2.2, we have $\lim\limits_{k \to +\infty} x_k = x^\star$.

## A.5 Proof of Corollary 4.1

Because $\tilde{\eta}_k \leq \eta_k$ and $\eta_k \leq c(k+1)^{-b}$, we have for $t \geq 1$ that

$$
\begin{aligned}
\theta_t &= \sum_{\ell=(t-1)(D+1)}^{t(D+1)-1} (2\tilde{\eta}_\ell(\|x_0 - x^\star\| + \eta_{\text{sum}})(1+\alpha L) + (\tilde{\eta}_\ell)^2) \\
&\leq \sum_{\ell=(t-1)(D+1)}^{t(D+1)-1} \frac{c}{(\ell+1)^b}(2(\|x_0 - x^\star\| + \eta_{\text{sum}})(1+\alpha L) + \frac{c}{(\ell+1)^b}) \\
&\leq \frac{c(D+1)}{((t-1)(D+1)+1)^b}(2(\|x_0 - x^\star\| + \eta_{\text{sum}})(1+\alpha L) + c).
\end{aligned}
$$

By the above inequality, we obtain that

$$
\begin{aligned}
\sum_{t=0}^{t_k-1} \rho^{t_k-t}\theta_t \leq &(2(\|x_0 - x^\star\| + \eta_{\text{sum}})(1+\alpha L) + c) \sum_{t=1}^{t_k-1} \rho^{t_k-t} \frac{c(D+1)}{((t-1)(D+1)+1)^b} \\
&+ \rho^{t_k}\theta_0
\end{aligned}
\tag{45}
$$

Let $p = t_k - t$. The right-hand side is rewritten as

$$
(2(\|x_0 - x^\star\| + \eta_{\text{sum}})(1+\alpha L) + c) \sum_{p=1}^{t_k-1} \rho^p \frac{c(D+1)}{((t_k-p-1)(D+1)+1)^b} + \rho^{t_k}\theta_0
$$

Firstly, when $0 \leq p \leq \lfloor(t_k-1)/2\rfloor$, we have that $t_k - 1 - p \geq (t_k-1)/2$, thereby having that

$$
\frac{1}{(t_k-1-p)^b} \leq \frac{2^b}{(t_k-1)^b}.
$$

Hence, we have that

$$
\begin{aligned}
\sum_{p=1}^{\lfloor(t_k-1)/2\rfloor} \rho^p \frac{c(D+1)}{((t_k-p-1)(D+1)+1)^b} &\leq \frac{2^b c}{(t_k-1)^b(D+1)^{b-1}} \sum_{p=1}^{\infty} \rho^p \\
&\leq \frac{2^b c\rho}{(t_k-1)^b(D+1)^{b-1}(1-\rho)}.
\end{aligned}
\tag{46}
$$

Secondly, when $\lfloor(t_k-1)/2\rfloor < p \leq t_k - 1$, we have that $\rho^p \leq \rho^{(t_k-1)/2}$. Here, we obtain that

$$
\sum_{p=\lfloor(t_k-1)/2\rfloor+1}^{t_k-1} \rho^p \frac{c(D+1)}{((t_k-p-1)(D+1)+1)^b} \leq c(D+1)(t_k-1)\rho^{(t_k-1)/2}.
\tag{47}
$$

Finally, we know that

$$
\begin{aligned}
\lim_{t_k \to \infty} \frac{2^b c(t_k-1)^b\rho}{(t_k-1)^b(D+1)^{b-1}(1-\rho)} &= \frac{2^b c\rho}{(D+1)^{b-1}(1-\rho)}, \\
\lim_{t_k \to \infty} c(D+1)(t_k-1)^{b+1}\rho^{(t_k-1)/2} = 0, \quad &\text{and} \quad \lim_{t_k \to \infty} \theta_0(t_k-1)^b\rho^{t_k} = 0.
\end{aligned}
$$

Therefore, by using inequalities (45), (46), and (47), we obtain (27). On the other hand, due to $\lim_{t_k \to \infty} \rho^{t_k}\|x_0 - x^\star\|(t_k-1)^b = 0$, and by using inequality (26), we have (28).

## B    Additional numerical experiments

In this section, firstly, we compare ADGM-AA with PIAG and DAve-G at the wall-clock time for logistic regression and least squares. Secondly, we report the robustness of ADGM-AA by varying the worker imbalance on the datasets Covtype and Cifar-10. We split samples of each dataset into two parts, with 25% and 75% of the samples, respectively. The first part is equally assigned to workers 1, 2,3, and 4, and the second to workers 5, 6, 7, and 8. Finally, we compare the test-set accuracy of ADGM-AA with that of PIAG and DAve-G on datasets Cifar-10 and Rcv1, where each dataset is split into training and test sets at a 7:3 ratio.

### B.1    Wall-clock time evaluation

At the wall-clock time, most experimental results show that ADGM-AA still converges faster than the two alternative methods DAve-G and PIAG in Figure 3 and 4. As $m$ increases from 6 to 16, the convergence of ADGM-AA still becomes faster for most results.

### B.2    Robustness to workload

In Figure 5, we present the robustness of ADGM-AA. From Figure 5, we see that even under varying levels of worker imbalance, ADGM-AA still converges faster than DAve-G and PIAG, and achieves the same high accuracy with respect to iterations and wall-clock time.

### B.3    Test accuracy

The experimental results are exhibited in Table 2 and 3 for the test-set accuracy. Here, all methods are implemented for the same number of iterations. We observe that the test-set accuracy of ADGM-AA is slightly better for the logistic regression and least squares. Together with the faster convergence reported above, these results show that ADGM-AA can accelerate the optimization process without degrading the generalization performance in these experiments.

Table 2: The test-set accuracy for logistic regression.

| Datasets | PIAG (%) | DAve-G (%) | ADGM-AA (%) |
|---|---|---|---|
| Cifar-10 | 69.69 | 71.75 | 73.00 |
| Rcv1 | 94.98 | 94.98 | 95.15 |

Table 3: The test-set accuracy for least squares.

| Datasets | PIAG (%) | DAve-G (%) | ADGM-AA (%) |
|---|---|---|---|
| Cifar-10 | 76.53 | 79.14 | 79.61 |
| Rcv1 | 97.49 | 97.54 | 97.58 |

## C    Diagnostic results for residual approximation

**Theoretical analysis.**    We first recall the acceleration mechanism suggested by the theory of Evans et al. (2020). For a standard fixed-point iteration

$$\tilde{x}_{k+1} = \mathrm{T}(x_k),$$

write the fixed-point residual at $x_k$ as

$$w_k = \mathrm{T}(x_k) - x_k = \tilde{x}_{k+1} - x_k.$$

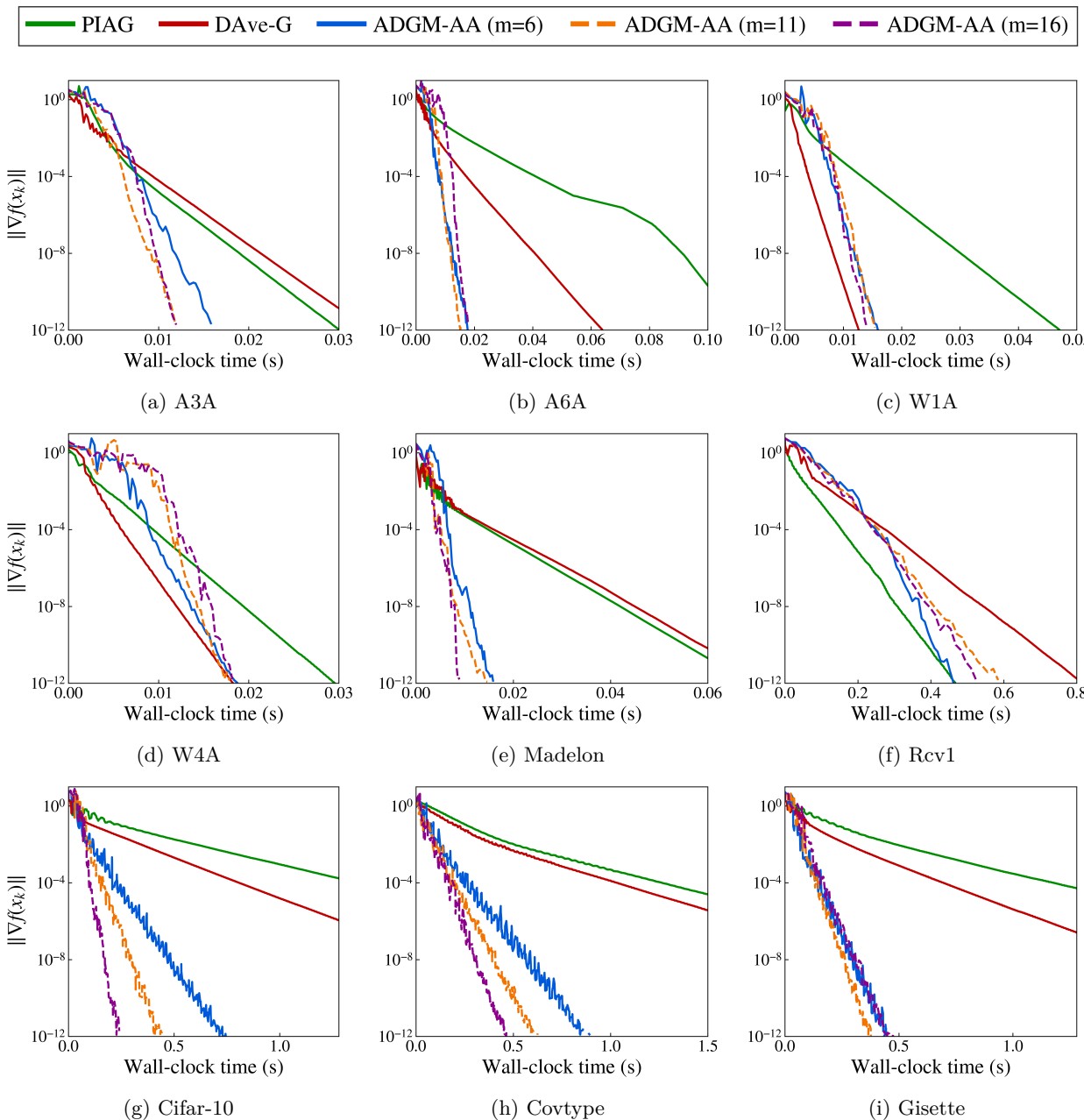

Figure 3: Comparison of PIAG, DAve-G, and ADGM-AA in solving logistic regression at the wall-clock time.

Adapting the notation to ours, the Anderson least-squares subproblem computes coefficients $\{\gamma_{t,k}\}_{t=0}^{m_k-1}$ over the recent residual window $\{w_{k-t}\}_{t=0}^{m_k-1}$ and forms the combined residual

$$w_k^\gamma = \sum_{t=0}^{m_k-1} \gamma_{t,k}\big(\mathrm{T}(x_{k-t}) - x_{k-t}\big).$$

Evans et al. (2020) define the stage gain $\theta_k$ by

$$\|w_k^\gamma\| = \theta_k \|w_k\|, \qquad 0 \le \theta_k \le 1,$$

which measures how much the Anderson least-squares subproblem reduces the residual relative to the unaccelerated fixed-point residual. Their analysis shows that, near a fixed point, Anderson acceleration improves

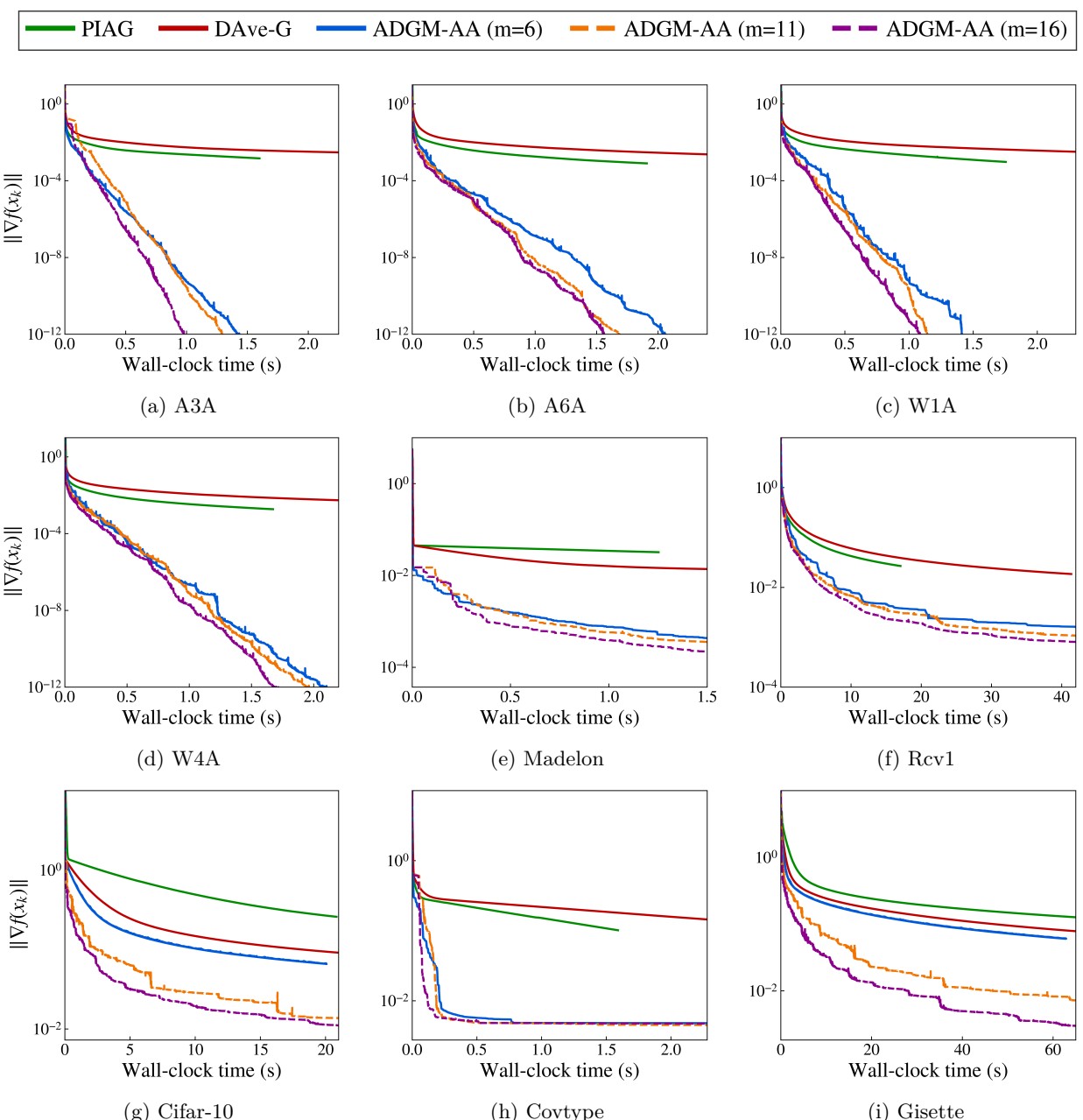

Figure 4: Comparison of PIAG, DAve-G, and ADGM-AA in solving least squares at the wall-clock time.

the first-order contraction of a linearly convergent fixed-point iteration by this gain, while additional higher-order terms are introduced. For intuition, the leading-order mechanism can be summarized as

$$\|w_{k+1}\| \leq \theta_k \kappa \|w_k\| + \sum_{j=0}^{m} \mathcal{O}\big(\|w_{k-j}\|^2\big), \tag{A}$$

where $\kappa$ is the contraction factor of the underlying fixed-point iteration. Thus, a small gain is most predictive of acceleration when the iterates are already in a local regime where the higher-order terms are relatively small. These higher-order terms can be interpreted as the non-affine modeling error underlying the Anderson residual approximation. In the special case of affine mappings, this modeling error vanishes and equation

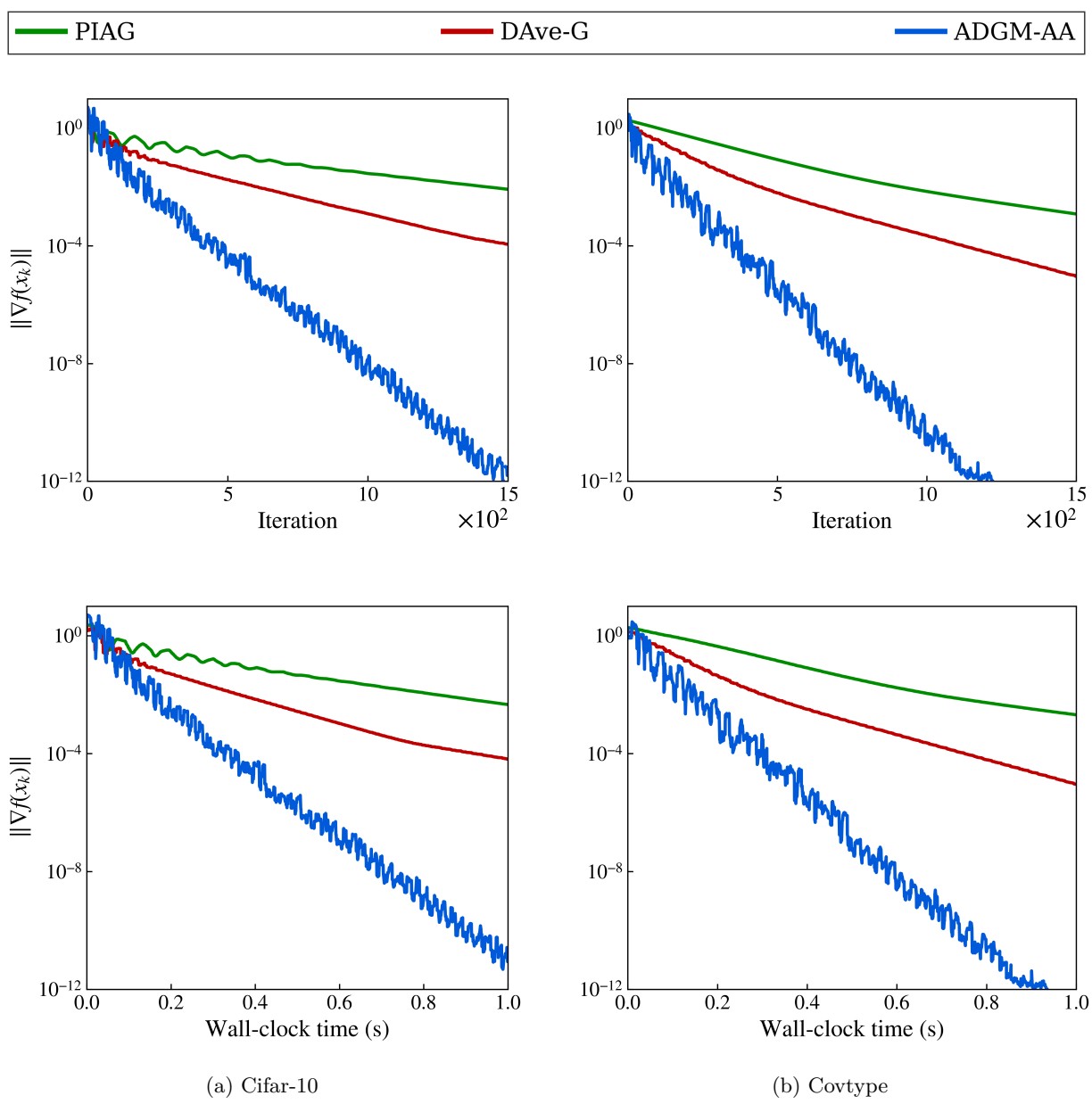

(a) Cifar-10

(b) Covtype

Figure 5: Comparison of PIAG, DAve-G, and ADGM-AA in solving logistic regression for the worker imbalance.

(A) can be simplified as

$$\|w_{k+1}\| \le \kappa\|w_k^\gamma\| = \theta_k\kappa\|w_k\|. \tag{B}$$

In the notation of ADGM-AA, the fixed-point map is our operator T, the analogue of $w_k$ is $\mathbf{r}_k$, and the analogue of $w_k^\gamma$ is $\mathbf{R}_k\Gamma_k$.

We emphasize that this result does not directly apply to our ADGM-AA setting. The analysis in Evans et al. (2020) is for standard fixed-point iterations under local smoothness and contraction assumptions, whereas ADGM-AA involves delayed residuals and a reference-path-based safe-guard. We therefore use this theory only as an interpretive guide for the observed behavior, not as a direct convergence theorem for ADGM-AA.

This interpretation suggests the following explanation for the transient behavior in logistic regression. Even when the Anderson least-squares subproblem produces a small predicted gain, i.e.,

$$\|\mathbf{R}_k\Gamma_k\|/\|\mathbf{r}_k\| < 1,$$

this only indicates that the residual has been reduced in the affine residual model used by AA. In the Evans-type residual expansion, this corresponds to reducing the first-order term, while additional higher-order terms remain. In the early stage, the residuals are still relatively large and the operator is not yet well approximated by a local affine model along the trajectory, so these higher-order, non-affine modeling errors may be non-negligible and can weaken the effect of the Anderson gain. As the iterates move into a local regime around the solution, which is guaranteed in our convergence analysis through the reference-path safeguard, the higher-order contribution becomes relatively smaller; consequently, the Anderson gain is more clearly reflected in the actual convergence.

**Numerical evidence.** To check the explanation above, we recorded the following diagnostic quantities during the ADGM-AA iterations. Let $\mathbf{r}(\mathbf{x}) = \mathrm{T}(\mathbf{x}) - \mathbf{x}$, $\mathbf{r}_k = \mathbf{r}(\hat{\mathbf{x}}_k)$, and

$$\mathbf{x}_{k+\frac{1}{2}} = \sum_{t=0}^{m_k-1} \gamma_{t,k}\hat{\mathbf{x}}_{k-t}, \qquad \mathbf{R}_k\Gamma_k = \sum_{t=0}^{m_k-1} \gamma_{t,k}\mathbf{r}_{k-t}.$$

In all normalized quantities, we set

$$\delta = 10^{-12}\max\{1, \|\mathbf{r}_0\|\},$$

which is used only to avoid division by zero. We first recorded the Evans-type predicted residual gain

$$\theta_k^{\mathrm{pred}} = \frac{\|\mathbf{R}_k\Gamma_k\|}{\|\mathbf{r}_k\| + \delta},$$

the non-affine residual-modeling error

$$E_k^{\mathrm{na}} = \frac{\left\|\mathbf{r}(\mathbf{x}_{k+\frac{1}{2}}) - \mathbf{R}_k\Gamma_k\right\|}{\|\mathbf{r}_k\| + \delta},$$

the true extrapolated residual gain

$$\theta_k^{\mathrm{half}} = \frac{\left\|\mathbf{r}(\mathbf{x}_{k+\frac{1}{2}})\right\|}{\|\mathbf{r}_k\| + \delta},$$

and the actual AA-step residual gain

$$\theta_k^{\mathrm{AA}} = \frac{\left\|\mathbf{r}(\mathbf{x}_{k+1}^{\mathrm{AA}})\right\|}{\|\mathbf{r}_k\| + \delta}.$$

Here, $\theta_k^{\mathrm{pred}}$ is the analogue of the stage gain in Evans et al. (2020); it is the predicted residual gain produced by the Anderson least-squares problem. The quantity $E_k^{\mathrm{na}}$, where "na" stands for non-affine, measures the discrepancy between the true residual at the extrapolated point and the affine residual model used in the Anderson least-squares problem. In contrast, $\theta_k^{\mathrm{half}}$ is the true residual gain at the intermediate point $\mathbf{x}_{k+\frac{1}{2}}$, and $\theta_k^{\mathrm{AA}}$ is the actual residual gain at the AA candidate $\mathbf{x}_{k+1}^{\mathrm{AA}}$. Moreover,

$$\theta_k^{\mathrm{half}} \le \theta_k^{\mathrm{pred}} + E_k^{\mathrm{na}},$$

so a small predicted gain is expected to translate into an actual residual reduction only when the non-affine modeling error is also sufficiently small.

The diagnostic results support this interpretation. For example, in the W1A logistic-regression diagnostic run shown in Figure 6, $E_k^{\mathrm{na}}$ is relatively large during the early iterations. During this phase, $\theta_k^{\mathrm{pred}}$ can already be smaller than one, but $\theta_k^{\mathrm{half}}$ and $\theta_k^{\mathrm{AA}}$ remain close to one or fluctuate. This means that the residual reduction predicted by the Anderson least-squares model is partially offset by the non-affine residual-modeling error,

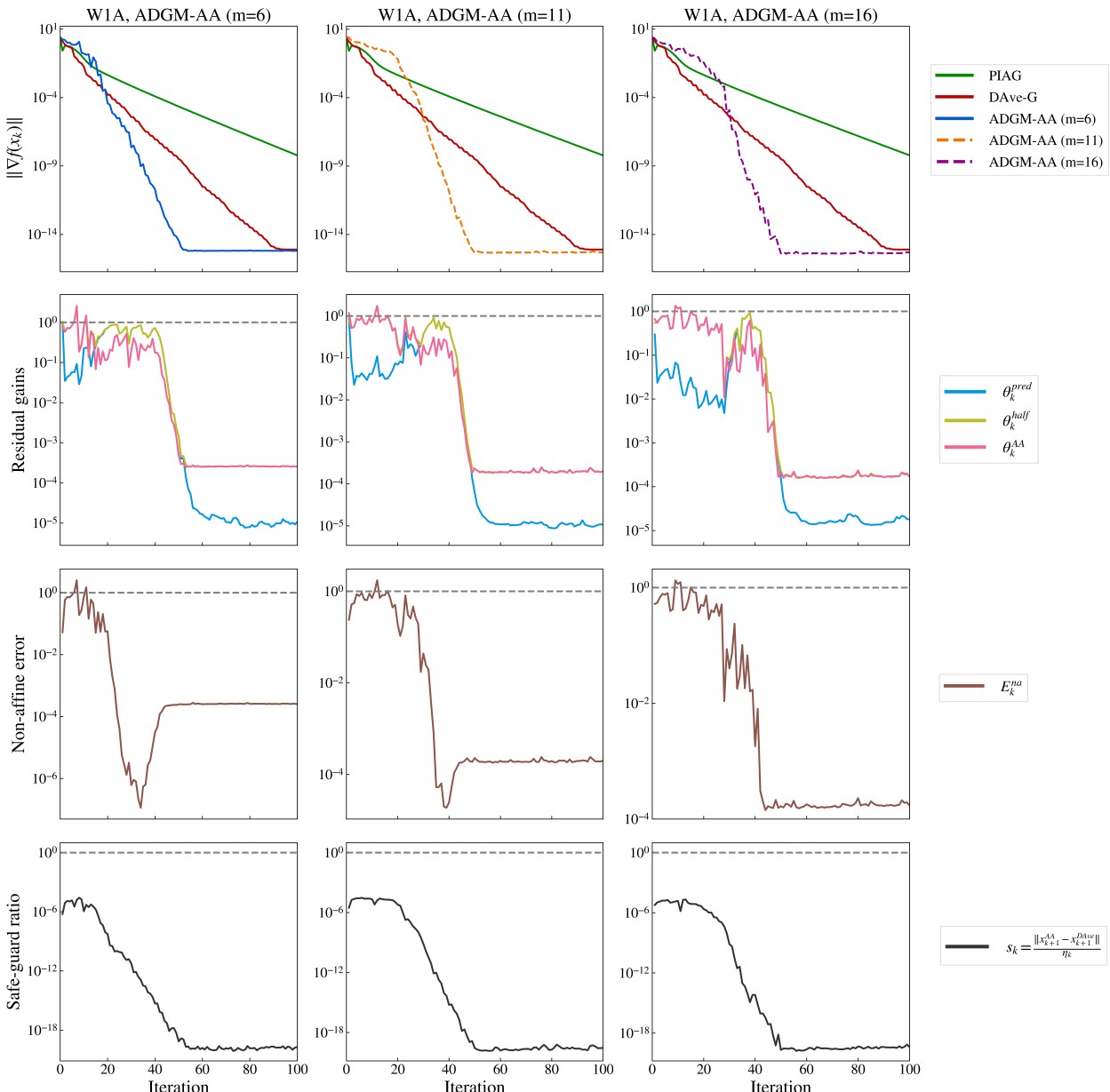

Figure 6: Diagnostic quantities for ADGM-AA on W1A logistic regression.

which is the counterpart of the higher-order contribution in the Evans-type residual expansion. After the transient phase, $E_k^{\mathrm{na}}$ decreases substantially, and both $\theta_k^{\mathrm{half}}$ and $\theta_k^{\mathrm{AA}}$ decrease accordingly. This coincides with the stage where ADGM-AA starts to outperform PIAG and DAve-G by a larger margin. Moreover, in these diagnostic runs, the safe-guard condition was always satisfied and the AA candidate was accepted. Hence, the delayed acceleration is not caused by safe-guard rejections, but by the fact that the predicted Anderson gain becomes effective only after the non-affine residual-modeling error has become small enough. As a consistency check, in the W1A least-squares diagnostic run shown in Figure 7, $E_k^{\mathrm{na}}$ remains close to numerical precision and the three residual gains nearly coincide.

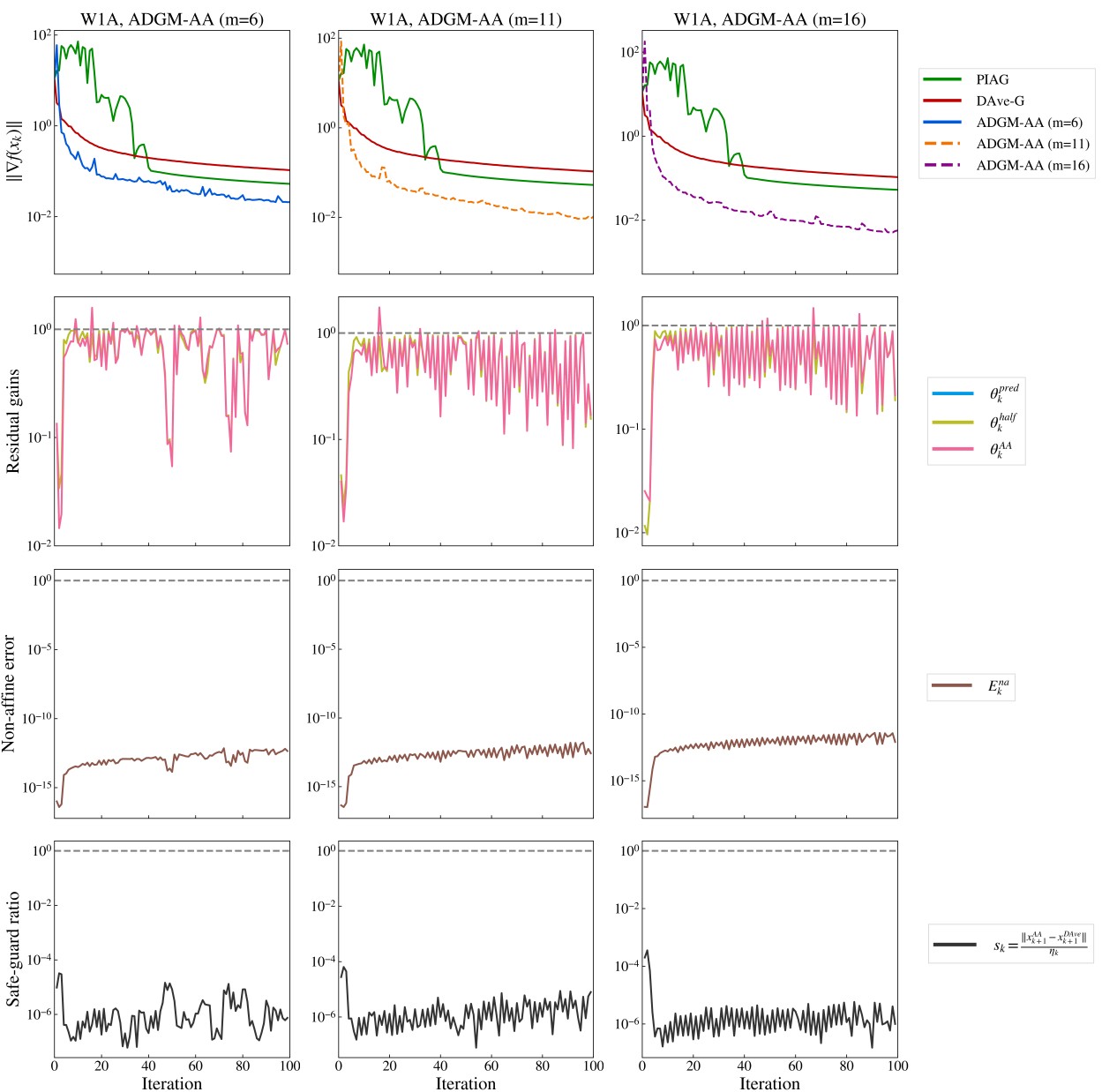

Figure 7: Diagnostic quantities for ADGM-AA on W1A least squares.

