# OpenReview forum: "Anderson Accelerated Asynchronous Method for Distributed Optimization"
_TMLR — Accepted by TMLR_

### Review · Reviewer_9BKM · 2026-05-24

**Summary Of Contributions:**

The paper proposes ADGM-AA, which applies Anderson acceleration (AA) to DAve-G, an asynchronous distributed gradient method over a master-worker architecture. The authors rewrite DAve-G as a fixed-point iteration so that the resulting algorithm can be used in an asynchronous and distributed way, propose a reference-path-based safeguard scheme that is distributively implementable, and prove the convergence of the algorithm. Experiments are conducted on logistic regression and least squares across nine datasets, showing fewer iterations to converge than PIAG and DAve-G. I have concerns about the gaps between the experimental evidence and the claims, detailed below.

**Additional Comments:**

The paper addresses an important problem with a well-structured framework. The concerns above are about experimental evidence rather than the core approach. Addressing them would improve overall clarity and make this a stronger submission.

**Audience:**

Yes

**Audience Explanation:**

Asynchronous Distributed Optimization is a well-studied and practically important problem, hence the scope of this paper is highly relevant to TMLR's audience.

**Broader Impact Concerns:**

No broader impact concerns.

**Claims And Evidence:**

No

**Claims Explanation:**

The paper's analysis is sound and the construction is well-motivated, however there are gaps between the claims and the evidence that need to be addressed.

1) The paper claims that AA "accelerates" DAve-G and that ADGM-AA "significantly outperforms" it. Authors haven't provided theoretical rate comparison between ADGM-AA vs DAve-G to support the claim, it is mostly empirical but that has another flaw.

2) The "outperforms" claim is also evaluated on iteration count alone, which is not a fair comparison. For example Hessian based methods have fast convergence vs iteration count but they have significant overhead in each iteration. Total time should be used for comparison rather than iteration count. ADGM-AA does extra work at the master each iteration (the least-squares solve and combination over the residual history), so fewer iterations does not imply faster convergence in practice.

3) Finally, the paper is framed as distributed machine learning but reports only the gradient norm on convex objectives (least squares and logistic regression), with no test-set loss or accuracy. The proofs and all experiments assume convex objective function, whereas the ML training the paper mentions is typically non-convex. The convex-only scope should be stated as an explicit limitation, and the paper should either evaluate generalizability or rescope the claims to convex optimization.

**Requested Changes:**

The authors should make the following changes:

1) Report wall-clock time and per-iteration cost. The "outperforms" claim is based on iteration counts only which is not a fair comparison.

2) Provide theoretical rate comparison between ADGM-AA vs Dave-G to support the claim that the ADGM-AA significantly outperforms the vanilla asynchronous distributed gradient method.

3) All datasets used are traditional ML classification tasks with convex objectives (least squares or logistic regression). Since the stated scope includes distributed ML, the paper should also evaluate generalizability on test datasets, or scope the claims to convex optimization.

---

> ### Author Response · Authors · 2026-06-20
> **Response to Reviewer 9BKM**
>
> We are grateful to the reviewer for the constructive feedback. Following the reviewer's comments, we have carefully revised the manuscript and provide our responses in the following OpenReview comments. Since the three weaknesses identified by the reviewer correspond exactly to the three requested changes, we focus our response on these requested changes.

---

> > ### Author Response · Authors · 2026-06-20
> > **Requested Change 1**
> >
> > Following the suggestion, we have reported the per-iteration cost and wall-clock-time results in the revised manuscript. The results show that ADGM-AA is faster in wall-clock time in most cases and comparable in the remaining cases. We have also added a short discussion of the extra AA overhead: the AA step is performed at the master and introduces no additional communication rounds, and the least-squares subproblem can be solved efficiently.

---

> > ### Author Response · Authors · 2026-06-20
> > **Requested Change 2, Existing AA Theory**
> >
> > We thank the reviewer for raising this important question. We agree that the theoretical acceleration of ADGM-AA over DAve-G should be stated carefully. In preparing this work, we surveyed the theory of Anderson acceleration (AA). Existing works do prove theoretical acceleration of AA in several settings, but the assumptions are substantially stronger than those used in our main result and the faster convergence is only proved locally in a neighborhood of the optimal solution for non-affine operators.
> >
> > For example, existing AA theory proves that AA improves the convergence rate of contractive fixed-point iterations in a neighborhood of a fixed point, but the result is local and contains additional higher-order terms (Evans et al., 2020). More recently, an improved root-linear convergence result has been proved for windowed AA, but it assumes an affine contractive fixed-point map $q(x)=Wx+a$, whose linear part $W$ is symmetric and satisfies $\|W\|<1$ (Garner et al., 2025). The extended results for non-affine operators assume a symmetric contractive Jacobian at the fixed point, use a modified AA subproblem, and include only improved local convergence. These contraction-type assumptions are not imposed in our main convex theorem.
> >
> > For an optimization-induced mapping such as $q(x)=x-\alpha\nabla f(x)$, smooth convexity generally gives nonexpansiveness rather than contraction, while contraction typically requires additional curvature, such as strong convexity, a positive definite Hessian, or nonsingularity of $A^\top A$ in least-squares problems.
> >
> > Moreover, ADGM-AA is not a standard synchronous fixed-point iteration. The baseline DAve-G is itself asynchronous and uses stale information returned by different workers. ADGM-AA constructs the Anderson step from delayed iterates, and different local gradients are generally not evaluated at the same global point. Therefore, the existing synchronous AA acceleration theory cannot be directly applied to prove a general theoretical acceleration result over the asynchronous DAve-G baseline.
> >
> > References:
> >
> > Evans et al. (2020). C. Evans, S. Pollock, L. G. Rebholz, and M. Xiao. A Proof That Anderson Acceleration Improves the Convergence Rate in Linearly Converging Fixed-Point Methods (But Not in Those Converging Quadratically). SIAM Journal on Numerical Analysis, 58(1):788-810, 2020. DOI: 10.1137/19M1245384.
> >
> > Garner et al. (2025). C. Garner, G. Lerman, and T. Zhang. Improved Convergence Factor of Windowed Anderson Acceleration for Symmetric Fixed-Point Iterations. arXiv preprint arXiv:2311.02490v3, 2025.

---

> > > ### Author Response · Authors · 2026-06-20
> > > **Requested Change 2, Scope of the Theoretical Claim**
> > >
> > > Our main theoretical contribution is instead to show that AA can be incorporated into asynchronous distributed optimization in a distributively implementable and globally convergent way. The first contribution is the incorporation of the AA scheme into the asynchronous distributed gradient method DAve-G, which is non-trivial because DAve-G is not naturally written as a fixed-point iteration. The second contribution is our reference-path-based safeguard scheme. Such a safeguard scheme is important because native AA is not globally convergent in general. A smooth strongly convex counterexample and a safeguard in the centralized proximal-gradient setting were given in (Mai & Johansson, 2020).
> > >
> > > In our master-worker setting, standard objective-decrease safeguards are not directly applicable, since the master does not in general have access to all local objectives $f_i$. We therefore use the DAve-G update as a reference path and accept the AA candidate only when its distance to the DAve-G reference update is controlled by a summable threshold sequence. The exact condition is stated in the revised manuscript. This condition is distributively checkable and enables us to prove global convergence of ADGM-AA with fixed step-sizes independent of the delay bound.
> > >
> > > We will revise the manuscript to make this distinction explicit. The theoretical claim is that ADGM-AA is a distributedly implementable AA-type method for asynchronous distributed optimization, equipped with a new reference-path-based safeguard and a global convergence guarantee under delay-independent fixed step-sizes. The statement that ADGM-AA significantly outperforms DAve-G will be presented as an empirical observation supported by our experiments. We believe that establishing a direct theoretical acceleration result for AA in the general convex asynchronous setting, especially relative to a strong delay-tolerant baseline such as DAve-G/DAve-RPG (Mishchenko et al., 2018), is an important and challenging direction for future work.
> > >
> > > References:
> > > Mai & Johansson (2020). Vien Mai and Mikael Johansson. Anderson Acceleration of Proximal Gradient Methods. In Proceedings of the 37th International Conference on Machine Learning, PMLR 119:6620-6629, 2020.
> > >
> > > Mishchenko et al. (2018). K. Mishchenko, F. Iutzeler, J. Malick, and M.-R. Amini. A Delay-tolerant Proximal-Gradient Algorithm for Distributed Learning. In Proceedings of the 35th International Conference on Machine Learning, PMLR 80:3587-3595, 2018.

---

> > ### Author Response · Authors · 2026-06-20
> > **Requested Change 3**
> >
> > Following the suggestion, we have clarified in the revised manuscript that the scope of the paper is convex distributed optimization, and that the experimental claims are restricted to the convex classification objectives considered in the manuscript. To address the reviewer's concern, we have also included test-accuracy results as supplementary experimental evidence, which show that ADGM-AA achieves test accuracy comparable to the baseline methods on the considered convex classification problems.

---

> > > ### Comment · Reviewer_9BKM · 2026-07-02
> > > **Thanks for the updates**
> > >
> > > The additional analysis has addressed all the issues and enhanced the paper's quality.

---

### Review · Reviewer_hMSo · 2026-05-25

**Summary Of Contributions:**

This paper mainly incorporated Anderson Acceleration (AA) into DAve-G, which first rewrote the update of DAve-G into a fixed-point iteration and then incorporated AA into the asynchronous distributed optimization setting. To ensure the global convergence of ADGM-AA for non-affine operators, this paper proposes a reference-path-based safeguard mechanism to restrict the iterate to be close to the reference path. The paper also provides the convergence analysis of the proposed ADGM-AA and presents the empirical improvements over DAve-G and PIAG.


Strength:

The paper applies Anderson Acceleration to asynchronous distributed optimization, which is rarely applied to distributed optimization or distributed machine learning.

Weakness:

The experimental analysis in Section 5 is relatively limited and needs more and deeper discussion of the observed behaviors.

**Audience:**

Yes

**Audience Explanation:**

The paper incorporates AA into asynchronous distributed optimization, which helps understand fixed-point acceleration methods. Also, the proposed safeguard mechanism is a novel mechanism to help asynchronous distributed optimization.

**Claims And Evidence:**

Yes

**Claims Explanation:**

The paper provides a clear theoretical analysis and clarifies the convergence guarantees under standard assumptions. The experimental results are consistent with the theoretical conclusions.

**Requested Changes:**

1. For the analysis of numerical experiments in Section 5, can authors provide a more detailed and comprehensive discussion of the observed convergence behaviors? I found that in Figure 1, several subfigures show that PIAG and DAve-G initially decrease faster than ADGM-AA during the early iterations, while ADGM-AA begins to converge faster only after a certain number of iterations. Can authors explain this observation? For example, is this behavior related to the initialization phase of Anderson acceleration, the quality of the residual approximation, or the safeguard mechanism limiting aggressive acceleration during early iterations?

2. In addition, the convergence behaviors in Figure 1 (logistic regression) and Figure 2 (least squares) appear noticeably different. Can authors provide further discussion connecting these empirical observations with the theoretical properties of Anderson acceleration in affine versus non-affine settings?

3. Also, can authors include the discussion of computational overhead introduced by the AA step relative to DAve-G?

4. Since the proposed method is asynchronous optimization, it would be helpful to evaluate the robustness of ADGM-AA under different cases, for example, can authors vary the worker imbalance to show the behaviors of the ADGM-AA?

---

> ### Author Response · Authors · 2026-06-20
> **Response to Reviewer hMSo**
>
> We sincerely thank the reviewer for the insightful comments, which helped us improve the clarity of the manuscript.

---

> > ### Author Response · Authors · 2026-06-20
> > **Requested Change 3**
> >
> > Thank you for the suggestion. We detail the additional computational overhead of the AA step relative to DAve-G here, and provide a brief version in Section 3 of the revised manuscript. To avoid overloading this OpenReview comment with long inline formulas, the notation follows the definitions in Section 3 of the revised manuscript.
> >
> > Let $m_k=\min\{m,k\}$, and let $S_k$ be the set of workers that return gradients at the $k$-th master update. Compared with DAve-G, ADGM-AA only adds the following master-side computations.
> >
> > 1. *Local copy update.* For each returned worker $i\in S_k$, the master records the point used by that worker and updates $\hat x_k^i=x_{t_i}$ and $g_k^i=\nabla f_i(x_{t_i})$. For $i\notin S_k$, the previous values are kept. This costs $O(|S_k|d)$.
> >
> > 2. *Residual construction and QR update.* After DAve-G computes its usual averaged update, ADGM-AA forms the full stacked residual by comparing this DAve-G update with each worker's delayed local copy. This residual lies in $\mathbb R^{nd}$, and constructing it costs $O(nd)$. The least-squares subproblem can then be implemented efficiently by maintaining the QR factorization of the residual matrix. As the AA window slides, the QR factors are updated by inserting the newest residual column and deleting the oldest one when the window is full. Following the QR-based AA implementation (Mai & Johansson, 2020), this maintenance costs $O(ndm_k+m_k^2)$. The exact residual formula and QR dimensions are given in Section 3 of the revised manuscript.
> >
> > 3. *AA coefficient computation.* Following the QR-based least-squares implementation (Mai & Johansson, 2020), the constrained least-squares problem is reduced to a small problem involving the triangular QR factor. The coefficient vector $\Gamma_k$ can then be computed by solving two triangular systems, which costs $O(m_k^2)$. The closed-form expression and triangular systems are given in Section 3 of the revised manuscript.
> >
> > 4. *AA candidate and safeguard.* The AA candidate is formed as a linear combination of recent DAve-G updates over the AA memory window, which costs $O(dm_k)$. The safeguard compares this candidate with the DAve-G reference update and costs $O(d)$. The exact candidate formula and safeguard condition are given in Section 3 of the revised manuscript.
> >
> > Therefore, the additional master-side computation per update is $O(|S_k|d+ndm_k+m_k^2+dm_k+d)$, dominated by $O(ndm_k)$ for small memory $m_k$. ADGM-AA introduces no additional worker-side gradient computation.
> >
> > References:
> >
> > Mai & Johansson (2020). Vien Mai and Mikael Johansson. Anderson acceleration of proximal gradient methods. In Proceedings of the 37th International Conference on Machine Learning, PMLR 119:6620-6629, 2020.

---

> > ### Author Response · Authors · 2026-06-20
> > **Requested Change 4**
> >
> > Following the suggestion, we have added robustness experiments under worker imbalance in the revised manuscript. In the new appendix experiment, we use the Covtype and Cifar-10 datasets and split the samples into two parts, containing 25% and 75% of the samples, respectively. The first part is equally assigned to workers 1, 2, 3, and 4, and the second part is equally assigned to workers 5, 6, 7, and 8. This creates an imbalanced worker workload while keeping the same master-worker implementation.
> >
> > The new results show that ADGM-AA still converges faster than DAve-G and PIAG under this worker-imbalance setting, both in terms of iterations and wall-clock time. These experiments provide additional evidence that the acceleration effect of ADGM-AA is robust to heterogeneous worker workloads.

---

> ### Author Response · Authors · 2026-06-20
> **Requested Change 1, Theoretical Analysis**
>
> We thank the reviewer for this insightful observation. We agree that, in several logistic-regression experiments in Figure 1, PIAG and DAve-G decrease $\|\nabla f(x_k)\|$ faster in the first few iterations, while ADGM-AA becomes substantially faster only after a transient phase. We believe that this behavior is consistent with the residual-based and local nature of Anderson acceleration, and provide both theoretical analysis and numerical evidence in the following. We have added this explanation to the revised Section 5.
>
> **Theoretical analysis.** We first recall the acceleration mechanism suggested by the theory of Anderson acceleration (Evans et al., 2020). For a standard fixed-point iteration with map $g$, write the fixed-point residual at $x_k$ as $w_k$.
>
> Adapting the notation to ours, the Anderson least-squares subproblem computes coefficients $\gamma_{t,k}$ over a recent residual window and forms a combined residual $w_k^\gamma$ as a linear combination of these residuals.
>
> The stage gain $\theta_k$ is defined in (Evans et al., 2020) by $\|w_k^\gamma\|=\theta_k\|w_k\|$, where $0\leq\theta_k\leq 1$, which measures how much the Anderson least-squares subproblem reduces the residual relative to the unaccelerated fixed-point residual. Their analysis shows that, near a fixed point, Anderson acceleration improves the first-order contraction of a linearly convergent fixed-point iteration by this gain, while additional higher-order terms are introduced.
>
> For intuition, the leading-order mechanism can be summarized as follows: the next residual is bounded by a first-order term scaled by the Anderson gain $\theta_k$ and the contraction factor $\kappa$, plus higher-order residual terms. The complete local bound is written explicitly in the revised Section 5.
>
> Thus, a small gain is most predictive of acceleration when the iterates are already in a local regime where the higher-order terms are relatively small. These higher-order terms can be interpreted as the non-affine modeling error underlying the Anderson residual approximation. In the special case of affine mappings, this modeling error vanishes, and the first-order Anderson gain directly controls the contraction. The corresponding simplified affine bound is also stated in the revised Section 5.
>
> In ADGM-AA, the fixed-point map $g$ in the standard fixed-point setting corresponds to our operator $T$. Using the notation of the manuscript, the analogue of $w_k$ is $r_k$, and the analogue of $w_k^\gamma$ is $R_k\Gamma_k$.
>
> We emphasize that this result does not directly apply to our ADGM-AA setting. The analysis in (Evans et al., 2020) is for standard fixed-point iterations under local smoothness and contraction assumptions, whereas ADGM-AA involves delayed residuals and a reference-path-based safeguard. We therefore use this theory only as an interpretive guide for the observed behavior, not as a direct convergence theorem for ADGM-AA.
>
> This interpretation suggests the following explanation for the transient behavior in logistic regression. Even when the Anderson least-squares subproblem produces a small predicted gain, i.e., $\|R_k\Gamma_k\|/\|r_k\|<1$, this only indicates that the residual has been reduced in the affine residual model used by AA. In the Evans-type residual expansion, this corresponds to reducing the first-order term, while additional higher-order terms remain. In the early stage, the residuals are still relatively large and the operator is not yet well approximated by a local affine model along the trajectory, so these higher-order, non-affine modeling errors may be non-negligible and can weaken the effect of the Anderson gain. As the iterates move into a local regime around the solution, which is guaranteed in our convergence analysis through the reference-path safeguard, the higher-order contribution becomes relatively smaller; consequently, the Anderson gain is more clearly reflected in the actual convergence.
>
> References:
>
> Evans et al. (2020). C. Evans, S. Pollock, L. G. Rebholz, and M. Xiao. A proof that Anderson acceleration improves the convergence rate in linearly converging fixed-point methods (but not in those converging quadratically). SIAM Journal on Numerical Analysis, 58(1):788-810, 2020. DOI: 10.1137/19M1245384.

---

> ### Author Response · Authors · 2026-06-20
> **Requested Change 1, Numerical Evidence**
>
> **Numerical evidence.** To check the explanation in revised Section 5, we recorded several diagnostic quantities during the ADGM-AA iterations. To avoid OpenReview rendering issues with long inline formulas, we only summarize the definitions here; the complete formulas and the corresponding plots are provided in the appendix section "Diagnostic results for residual approximation" of the revised manuscript. Let $r(\mathbf{x})$ denote the fixed-point residual induced by $T$, and let $r_k$ be the current residual. We also use an intermediate extrapolated point and the Anderson linear residual model $R_k\Gamma_k$, as defined in the manuscript.
>
> In all normalized quantities, we set $\delta=10^{-12}\max\{1,\|r_0\|\}$, which is used only to avoid division by zero. We recorded four quantities: the predicted residual gain $\theta_k^{\mathrm{pred}}$, the non-affine residual-modeling error $E_k^{\mathrm{na}}$, the true extrapolated residual gain $\theta_k^{\mathrm{half}}$, and the actual AA-step residual gain $\theta_k^{\mathrm{AA}}$.
>
> Here, $\theta_k^{\mathrm{pred}}$ is the analogue of the stage gain in (Evans et al., 2020); it measures the residual reduction predicted by the Anderson least-squares problem. The quantity $E_k^{\mathrm{na}}$, where "na" stands for non-affine, measures the discrepancy between the true residual at the extrapolated point and the affine residual model used in the Anderson least-squares problem. In contrast, $\theta_k^{\mathrm{half}}$ is the true residual gain at the intermediate point, and $\theta_k^{\mathrm{AA}}$ is the actual residual gain at the AA candidate. These quantities are all normalized by $\|r_k\|+\delta$. A small predicted gain is therefore expected to translate into an actual residual reduction only when the non-affine modeling error is also sufficiently small.
>
> The diagnostic results support this interpretation. For example, in the W1A logistic-regression diagnostic run, $E_k^{\mathrm{na}}$ is relatively large during the early iterations. During this phase, $\theta_k^{\mathrm{pred}}$ can already be smaller than one, but $\theta_k^{\mathrm{half}}$ and $\theta_k^{\mathrm{AA}}$ remain close to one or fluctuate. This means that the residual reduction predicted by the Anderson least-squares model is partially offset by the non-affine residual-modeling error, which is the counterpart of the higher-order contribution in the Evans-type residual expansion.
>
> After the transient phase, $E_k^{\mathrm{na}}$ decreases substantially, and both $\theta_k^{\mathrm{half}}$ and $\theta_k^{\mathrm{AA}}$ decrease accordingly. This coincides with the stage where ADGM-AA starts to outperform PIAG and DAve-G by a larger margin. We have included the W1A diagnostic plots in the appendix section "Diagnostic results for residual approximation" of the revised manuscript, together with the corresponding W1A least-squares diagnostic plots as a control case. Moreover, in these diagnostic runs, the safeguard condition was always satisfied and the AA candidate was accepted. Hence, the delayed acceleration is not caused by safeguard rejections, but by the fact that the predicted Anderson gain becomes effective only after the non-affine residual-modeling error has become small enough. As a consistency check, in the W1A least-squares diagnostic run, $E_k^{\mathrm{na}}$ remains close to numerical precision and the three residual gains nearly coincide.
>
> References:
> Evans et al. (2020). C. Evans, S. Pollock, L. G. Rebholz, and M. Xiao. A proof that Anderson acceleration improves the convergence rate in linearly converging fixed-point methods (but not in those converging quadratically). SIAM Journal on Numerical Analysis, 58(1):788-810, 2020. DOI: 10.1137/19M1245384.

---

> ### Author Response · Authors · 2026-06-20
> **Requested Change 2**
>
> We also thank the reviewer for pointing out the different behaviors between the logistic-regression experiments in Figure 1 and the least-squares experiments in Figure 2: ADGM-AA becomes substantially faster only after a transient phase for logistic regression, but the convergence speed does not vary significantly across iterations for least squares. Using the diagnostic viewpoint described in revised Section 5 and in the appendix section "Diagnostic results for residual approximation", the main reason for the difference is whether the fixed-point operator $T$ is affine.
>
> For least squares, the objective is quadratic, $\nabla F$ is affine, and the fixed-point operator $T$ is affine because $P_C$ is linear. Therefore, under the standard Anderson affine-combination constraint, the residual model $R_k\Gamma_k$ is exact up to numerical precision. This explains why, in the least-squares diagnostics, $E_k^{\mathrm{na}}$ stays close to zero and the predicted, extrapolated, and AA-step residual gains nearly coincide. The explicit form of $T$ and the residual model are given in the revised manuscript.
>
> For logistic regression, the non-affine operator gives rise to higher-order terms in the local AA bound stated in the revised Section 5. These terms can weaken the effect of the predicted Anderson gain during the early iterations. In the affine least-squares case, the higher-order modeling error vanishes, and the bound reduces to a first-order contraction controlled by the Anderson gain. Therefore, the difference between Figure 1 and Figure 2 is explained by the affine versus non-affine nature of $T$. In least squares, the Anderson residual approximation is essentially exact, while in logistic regression it becomes reliable only after the iterates enter a local regime.

---

### Review · Reviewer_zoLH · 2026-06-06

**Summary Of Contributions:**

The paper proposes a new algorithm for asynchronous distributed optimization based on Anderson acceleration. The convergence of the proposed method is established under standard assumptions. Experimental results on logistic regression and least-squares problems demonstrate significant speedups compared with widely used benchmark methods.

**Audience:**

Yes

**Audience Explanation:**

The application of Anderson acceleration to asynchronous distributed optimization is of clear practical relevance. The algorithmic contributions are of interest to the optimization communities, and the reported iteration-count improvements are significant.

**Broader Impact Concerns:**

The paper poses no foreseeable ethical concerns or broader societal risks that would require a Broader Impact Statement.

**Claims And Evidence:**

Yes

**Claims Explanation:**

The submission's claims are supported and empirical comparisons show iteration-count improvements.

**Requested Changes:**

1. The paper studies problem (1) but does not explicitly state that all local objective functions $f_i$ are assumed convex. Since the convergence analysis in Section 4 relies on convexity, this assumption should be added explicitly as a standing assumption, alongside Assumptions 4.1--4.3.

2. Let $\mathbf{x} = ((x^1)^T, \ldots, (x^n)^T)^T \in \mathbb{R}^{nd}$. The projection $P_{\mathcal{C}}(\mathbf{x})$ yields a vector whose $n$ blocks are all equal to the mean $\frac{1}{n}\sum_{i=1}^n x^i \in \mathbb{R}^d$. It follows that least square problem (15) can be formulated in $\mathbb{R}^d$ rather than $\mathbb{R}^{nd}$. The authors should make this reduction explicit, update Algorithm 1 to reflect that only $d$-dimensional quantities are stored and communicated, and discuss the resulting reduction in memory and computational cost.

3. $S_k$ is not defined in Algorithm 1.

---

> ### Author Response · Authors · 2026-06-20
> **Response to Reviewer zoLH**
>
> We would like to thank the reviewer for the positive comments and helpful suggestions.
>
> **Response to requested change 1.** Following the suggestion, we have stated that "each $f_i$ is convex" in Assumption 4.1 of the revised manuscript.
>
> **Response to requested change 2.** Thank you for the comment. We agree that $P_C(\mathbf{x})$ lies in the consensus subspace and can therefore be represented by a $d$-dimensional vector.
>
> Then, the AA candidate can be interpreted as a linear combination of recent DAve-G iterates, with coefficients $\gamma_{t,k}$ over the AA memory window. Each DAve-G iterate is obtained by averaging the corresponding delayed local gradient steps over all workers. The detailed formula is given in Section 3 of the revised manuscript.
>
> However, the least-squares problem (15) in the revised manuscript cannot generally be reduced to $\mathbb R^d$. In particular, problem (15) minimizes $\|\mathbf{R}_k\Gamma\|$ over $\Gamma\in\mathbb{R}^m$ with $\Gamma^T\mathbf{1}=1$, and the optimizer is $\Gamma_k$.
>
> Here, $\mathbf{R}_k$ is the stacked residual matrix over the AA memory window. Each residual vector $\mathbf{r}_k$ consists of worker-level blocks, and the block for worker $i$ is the difference between the DAve-G update and the delayed local copy held for that worker. These definitions are stated explicitly in the manuscript.
>
> Note that these blocks $r_k^i$ are generally different because the delayed local copies $\hat x_k^i$ are different in the asynchronous setting. Hence, in general problem (15) is not equivalent to a $d$-dimensional least-squares problem.
>
> Following the suggestion, we have discussed the computation, communication, and memory costs in the revised manuscript, and include the relevant text here for your convenience.
>
> *Computation, communication, and memory costs.* ADGM-AA follows the same master-worker communication pattern as DAve-G. At the $k$th master update, each worker in $S_k$ sends a $d$-dimensional gradient vector together with its iteration index to the master, and the master sends a $d$-dimensional iterate together with its iteration index back to these workers. Hence, the per-worker memory and communication cost remains $O(d+1)$. Since the AA step is performed entirely at the master, ADGM-AA incurs essentially no additional worker-side computation compared with DAve-G; the only extra worker-side requirement is storing and communicating a scalar iteration index.
>
> The additional overhead of ADGM-AA comes from the master-side AA step, whose dominant cost is solving problem (15). Let $m_k=\min\{m,k\}$. Following the QR-based AA implementation (Mai & Johansson, 2020), solving problem (15) costs at most $O(m_k^2+ndm_k)$ per iteration, while the corresponding memory cost is $O(ndm_k+m_k^2+dm_k)$. The total communication cost per iteration is $O(|S_k|(d+1))$, due to transmitting $(x_k,k)$ from the master to the workers in $S_k$ and receiving their gradient vectors with iteration indices. This is only slightly higher than the $O(|S_k|d)$ communication cost of DAve-G.
>
> In summary, although ADGM-AA does not reduce the least-squares problem to $\mathbb{R}^d$, this subproblem can still be handled efficiently at the master. At the same time, ADGM-AA keeps both the worker communication and the final iterate representation $d$-dimensional. Compared with DAve-G, ADGM-AA introduces almost no additional communication cost, no additional worker-side computation, and only extra computation and memory costs at the master. Numerical experiments in Section 5 show that these additional master-side costs are worthwhile, as ADGM-AA achieves much faster convergence than DAve-G.
>
> **Response to requested change 3.** Thank you for your kind reminder. In the revised manuscript, we clarified that $S_k$ is the set of workers the master node receives messages from at the $k$th iteration and is not fixed in advance.
>
> References:
>
> Mai & Johansson (2020). V. V. Mai and M. Johansson. Anderson acceleration of proximal gradient methods. In Proceedings of the 37th International Conference on Machine Learning, PMLR 119:6620-6629, 2020.

---

### Decision · Action_Editor_cU2t · 2026-07-07

**Recommendation:** Accept as is

**Audience:**

Yes

**Audience Explanation:**

Asynchronous distributed optimization is an important topic for large-scale learning.

**Claims And Evidence:**

Yes

**Claims Explanation:**

The paper investigates Anderson acceleration in asynchronous distributed gradient methods. Proofs of superior performance are difficult to obtain theoretically nevertheless, the authors still show convergence proofs and rates. The various numerical illustrations provide evidence that this type of acceleration can have a positive impact on convergence.